# On the Query Complexity of Verifier-Assisted Language Generation

**Edoardo Botta** [* 1]  **Yuchen Li** [* 1]  **Aashay Mehta** [* 1]  **Jordan T. Ash** [2]  **Cyril Zhang** [2]  **Andrej Risteski** [1]

## Abstract

Recently, a plethora of works have proposed inference-time algorithms (e.g. best-of-n), which incorporate verifiers to assist the generation process. Their quality-efficiency trade-offs have been empirically benchmarked on a variety of constrained generation tasks, but the algorithmic design landscape is still largely poorly understood. In this paper, we develop a mathematical framework for reasoning about constrained generation using a pre-trained language model generator oracle and a process verifier—which can decide whether a prefix can be extended to a string which satisfies the constraints of choice. We show that even in very simple settings, access to a verifier can render an intractable problem (information-theoretically or computationally) to a tractable one. In fact, we show even simple algorithms, like tokenwise rejection sampling, can enjoy significant benefits from access to a verifier. Empirically, we show that a natural modification of tokenwise rejection sampling, in which the sampler is allowed to "backtrack" (i.e., erase the final few generated tokens) has robust and substantive benefits over natural baselines (e.g. (blockwise) rejection sampling, nucleus sampling)—both in terms of computational efficiency, accuracy and diversity. [1]

## 1. Introduction

The fast-evolving area of inference-time algorithms concerns itself with leveraging the already-impressive capabilities of language models (Raffel et al., 2020; Brown et al., 2020; Touvron et al., 2023), together with a *verifier* which can score generations of the language model. In the sim-

plest form, called *best-of-N*, the language model generates $N$ candidate responses, which are then scored by the verifier, and the highest-scored candidate response is chosen as the output of the inference process (Cobbe et al., 2021; Nakano et al., 2022). If the verifier can score partial generations (sometimes called *process reward*), the space for inference-time algorithms gets much richer: e.g., the final answer can be generated incrementally, using the verifier to guide the process (e.g., by incremental (blockwise) best-of-N, or more complicated strategies like Monte-Carlo-Tree-Search (Browne et al., 2012; Hao et al., 2023)). Importantly, though a flurry of recent papers consider "scaling laws" of natural strategies, the algorithm design space of verifier-aided inference-time algorithms is still opaque. In particular, the *value of a verifier*—and the *relationship it needs to have to the generator* is not well understood.

In this paper, we show that a good verifier can substantially (both in theory and in practice) decrease the computational cost of natural generation tasks, using a pre-trained language model as an *oracle*. In particular, we show that:

- Even simple *constrained generation tasks*—where we are trying to generate a string in the support of a language oracle, subject to some structural constraint (e.g. describable as a simple formal language, like a regular language)—can be *computationally intractable in the absence of a verifier*.

- Conversely, access to a good *process verifier*, one that can decide whether a prefix can be completed to a constraint-satisfying string, can remove these intractabilities. Moreover, even simple algorithms like tokenwise rejection sampling—wherein we generate the string one token at a time, using the process verifier as a means to accept or reject—can have substantive computational benefits over the baseline of rejection sampling.

- Finally, on natural constrained generation tasks—namely, generating test cases for Python functions with a pretrained CodeLlama (Roziere et al., 2023)—a *verifier can be trained*, such that a simple, but natural generalization of tokenwise rejection sampling which is allowed to "backtrack" the last few generated tokens, achieves substantial benefits in computational efficiency, accuracy, and diversity of the generations.

---

[*]Equal contribution  [1]Carnegie Mellon University  [2]Microsoft Research NYC. Correspondence to: Yuchen Li <yuchenl4@cs.cmu.edu>, Andrej Risteski <aristesk@andrew.cmu.edu>.

*Proceedings of the 42$^{nd}$ International Conference on Machine Learning*, Vancouver, Canada. PMLR 267, 2025. Copyright 2025 by the author(s).

[1]Our codes are released at https://github.com/YuchenLi01/LM_Query_Complexity

## 2. Setup and notation

Throughout, we let $\Sigma$ be a nonempty finite set, denoting the vocabulary. We denote as $\Sigma^i$ the set of strings of length $i$ and by $\Sigma^* = \cup_{i \in \mathbb{N}} \Sigma^i$ the set of all finite strings on $\Sigma$. Given a string $s \in \Sigma^*$, we denote as $s_i$ its $i$-th element and as $s_{i:j}$ the substring of $s$ starting at its $i$-element and ending at its $j$-element, included. We use $|s|$ to denote the length of string $s$ and $\epsilon$ to denote the empty string. Finally, we let $x \circ y$ denote the concatenation of string $x$ followed by string $y$.

**Definition 1** (Autoregressive oracle). *An **autoregressive oracle** $\mathcal{O}$ takes as input a string $s \in \Sigma^*$ and returns a sample from a next-token distribution $\mathcal{O}(s) : \Sigma \to \mathbb{R}^+$.*

*We will denote the corresponding joint distribution over strings $s \in \Sigma^*$ as $p_{\mathcal{O}} : \Sigma^* \to \mathbb{R}^+$. Correspondingly, $\forall s \in \Sigma^*$, let $p_{\mathcal{O}}(\cdot \mid s)$ denote the distribution over completions of $s$ predicted by $\mathcal{O}$.*

**Definition 2** (Constrained generation). *Constrained generation with respect to an oracle $\mathcal{O}$, a constraint set $A$, and vocabulary $\Sigma$ is the task of producing an element $s \in A \subseteq \Sigma^*$ such that $p_{\mathcal{O}}(s) > 0$. If no such $s$ exists, the algorithm needs to output* `FAIL`.

When not clear from context, we will specify instances of this task by the triple $(\Sigma, A, \mathcal{O})$. Under suitable choices of the vocabulary $\Sigma$ and the target domain $A$, one recovers several language modeling tasks of theoretical and practical relevance as special cases of constrained generation. Specifically, our experiments consider the tasks of generating (i) valid strings under the Dyck grammar (Section 5.1) and (ii) valid test cases for a given Python functions (Section 5.2), where the oracles return samples from an appropriately pretrained language model. We recover these tasks from Definition 2 by setting:

- (i) $\Sigma$ as the set of open and close parentheses, and $A$ as the set of valid sequences of given length.

- (ii) $\Sigma$ as a set of characters from the Unicode standard (possibly after tokenization) and $A$ as the set of strings that are valid test cases for an input function in the Python programming language.

Note that this task is easier than the task of sampling according to the *restricted distribution* $p(s) \propto \mathbf{1}(s \in A)p_{\mathcal{O}}(s)$, which asks that the relative weights of the strings $s \in A$ that are generated match the probabilities assigned by $p_{\mathcal{O}}$. However, in many settings—e.g., generating a proof of a mathematical problem, or code that performs some intended functionality—we merely care about producing one good sample.

We will be considering "process verifiers" that take as input a prefix $s$, and output whether or not such a prefix can be completed to a string $s \circ s' \in A$. This is a natural formalization of a "process reward", as it assigns a belief to a partial generation. In the theoretical results (Section 3 and 4), we'll assume access to such an idealized verifier. In the empirical results (Section 5), such a verifier will be trained and will output a value between 0 and 1, which can be naturally interpreted as a probability that the prefix $s$ is completable to a string $s \circ s' \in A$.

**Definition 3** (Process verifier). *Given a constraint set $A$, a verifier is a function $V : \Sigma^* \to \{0, 1\}$ such that $\forall s \in \Sigma^*$, $V(s) = 1$ if and only if $\exists s' \in \Sigma^*$ such that $s \circ s' \in A$.*

Designing algorithms given access to oracles which perform certain tasks, is a classical tool in computer science (this is the basis of Turing reductions in computational complexity), as well as optimization (e.g., zero-order optimization assumes a value oracle for a function, first-order optimization a gradient oracle, etc.) In the context of generative modeling, analyses based on oracle complexity have been carried out in the settings of diffusion models, where sampling algorithms rely on score oracles (Chen et al., 2022).

We will consider several natural algorithms that use an autoregressive oracle and a (process) verifier:

**Definition 4** (Rejection sampling). *Rejection sampling works by repeatedly generating a string $s$ according to $p_{\mathcal{O}}$, then running a verifier $V$ on the complete string—and accepting when the verifier outputs $V(s) = 1$.*

Note, this algorithm only needs a verifier that decides the membership in $A$, rather than a process verifier. On the other hand, because the entire string needs to be generated first before being verified—the number of generations until the verifier accepts is likely very large.

**Definition 5** (Tokenwise rejection sampling). *Tokenwise rejection sampling works by generating a string one token at a time. To generate the next token $t$, given a prefix $s$, we sample $t \sim \mathcal{O}(s)$, and run the process verifier on $V(s \circ t)$. We repeat this, until $V(s \circ t) = 1$, then proceed to the next token.*

This algorithm requires a process verifier. However, since a partial string is accepted only if the process verifier accepts, the number of generations needed is likely to be smaller. In fact, we provide a very simple example in Section 4.

Finally, we consider a "backtracking" strategy, in which the model is allowed to erase some of its generations. The reasons to consider such a strategy is to allow the model to get "unstuck": if the process verifier decides the current prefix cannot be completed to a valid string in $A$, it is possible that erasing the last few tokens will make it easier for the model to correct its mistake, compared to erasing just the last token. More formally, the framework of our algorithm

is given by Algorithm 1 below. [2]

---
**Algorithm 1** Tokenwise rejection sampling with backtracking

---
1: **Input:** Prompt $x$, generator $\mathcal{O}$, verifier $\mathsf{V}$, length $D \in \mathbb{N}_+$, backtrack quota $Q \in \mathbb{N}$, backtrack stride $B \in \mathbb{N}_+$
2: $s \leftarrow \epsilon$
3: **while** $|s| < D$ and $s_{|s|} \neq \texttt{<eos>}$ **do**
4:      Sample $\hat{s} \sim \mathcal{O}(x \circ s)$
5:      $s \leftarrow s \circ \hat{s}$
6:      **if** $Q > 0$ and $\mathsf{V}(x \circ s) = 0$ **then**
7:          $s \leftarrow s_{1:|s|-B}$
8:          $Q \leftarrow Q - 1$
9:          **for** i in $1 \cdots B$ **do**
10:              Choose $\hat{s} \in \arg\max \mathcal{O}(x \circ s)$
11:              $s \leftarrow s \circ \hat{s}$
12:          **end for**
13:      **end if**
14: **end while**

---

When arguing about lower bounds, a natural lower bound on the complexity of an algorithm is the number of oracle calls needed[3], particularly so when this dominates the cost of the algorithm, as is frequently the case for language models:

**Definition 6** (Oracle complexity). *Given a (possibly randomized) algorithm $\mathcal{A}$ that solves the constrained generation instance $(\Sigma, A, \mathcal{O})$, the **oracle complexity** of $\mathcal{A}$ is defined as the expected number of calls to the oracle made by $\mathcal{A}$ to solve $(\Sigma, A, \mathcal{O})$, namely:*

$$\mathcal{C}(\mathcal{A}) = \mathbb{E}[\#\text{calls to } \mathcal{O} \text{ made by running } \mathcal{A}],$$

*where the expectation is taken over the randomness of the oracle $\mathcal{O}$ and the randomness of the algorithm $\mathcal{A}$.*

Finally, we recall the classical knapsack problem, which will be used in a reduction to prove computational intractability results for the constrained generation task:

**Definition 7** (Knapsack problem). *Given a set of weights $\{X_i \in \mathbb{Z}_{\geq 0} \mid i \in [D]\}$ and $c \in \mathbb{Z}_{\geq 0}$, the knapsack problem*

---

[2]The algorithm is a bit more involved, so we will describe it in pseudocode rather than text. Besides the notations in Section 2, Algorithm 1 uses the following additional common conventions: $\texttt{<eos>}$ denotes the end-of-sequence token; $s_{|s|} \neq \texttt{<eos>}$ is understood as True when $s = \varepsilon$; for any starting index $i$ and ending index $j$, if $i > j$, then $s_{i:j} = \varepsilon$. In line 10, why redoing the erased positions using argmax: our results in Section 5.1.1 suggests that *out-of-distribution prefix* is a cause of generator mistakes. As a remedy, redoing the erased positions using argmax is intended to increase the generator-predicted probability of the currently sampled prefix. We include an ablation study in Appendix B.3 verifying that this improves the accuracy.

[3]In our case, the number of calls is a randomized quantity, so a natural quantity to consider is the expected number of oracle calls. It is of course reasonable to consider finer-grained notions like tail bounds on the number of calls.

*seeks an assignment of the variables $(a_i)_{i=1}^D$, with $a_i \in \{0, 1\} \, \forall i \in [D]$ such that $c = \sum_{i=1}^D a_i X_i$.*

The problem is (weakly) NP-hard, even for some very special choices of $c, X_i$.

# 3. Constrained generation is hard without a verifier

First, we show that the constrained generation task (Definition 2), without access to a process verifier can be intractable—even if the constraint set $A$ is extremely simple (e.g. the parity of a binary string).

The source of intractability can be *information-theoretic*: namely, if the oracle does not have a succinct description, the algorithm may need to query it prohibitively many times to identify what oracle it's interacting with. We view this as a plausible obstruction in practice as well: language models frequently behave unpredictably "in-the-tails", which becomes increasingly more likely when generating long strings. Thus, to inspect the behavior of the model on long strings, many queries are needed.

The source of the intractability can also be *computational*: namely, even if the oracle is very simple (e.g., a uniform distribution), generating a member of $A$ can be NP-hard, even if checking membership in $A$ can be done efficiently. Perhaps this should not come as a surprise: after all, easy verification of membership, but hard generation, is the hallmark of NP-hard problems.

Proceeding to the first result, we show the following:

**Theorem 1.** *There exists a constrained generation task $(\Sigma, A, \mathcal{O})$ for which $\Sigma = \{0, 1\}$, $A \subseteq \Sigma^D$, and $\mathcal{O}$ is an (unknown) member of a set of $2^{D-1}$ possible oracles, such that any (possibly randomized) algorithm $\mathcal{A}$ has an (expected) oracle complexity of at least $2^{D-1}$.*

Intuitively, the lower bound is shown by engineering a scenario such that the behavior of the oracle on long strings is unknown to the algorithm—but success of the generation task relies on "guessing" this behavior correctly.

*Proof.* Consider the constrained generation task $(\Sigma, A, \mathcal{O}_{\hat{s}})$, such that $\Sigma := \{0, 1\}$, $A := \{s \in \Sigma^D : \sum_{i=1}^D s_i \mod 2 = 0\}$ for some fixed $D \in \mathbb{Z}_+$. Moreover, the oracle $\mathcal{O}_{\hat{s}}$ is indexed by an (unknown to the algorithm) $\hat{s} \in \Sigma^{D-1}$, and it specifies the autoregressive distribution defined s.t. $\forall s \in \Sigma^*, |s| < D - 1$, we have $p_{\mathcal{O}_{\hat{s}}}(1|s) = p_{\mathcal{O}_{\hat{s}}}(0|s) = 1/2$; while for $s \in \Sigma^*, |s| = D - 1$, it satisfies:

$\forall s \neq \hat{s} \in \Sigma^{D-1}, s_D \in \{0, 1\}$, we have:

$$p_{\mathcal{O}_{\hat{s}}}(s_D \mid s) = \begin{cases} 1, & \text{if } \left(\sum_{j=1}^{D-1} s_j + s_D\right) \mod 2 = 1 \\ 0, & \text{otherwise} \end{cases}$$

(1)

For $s = \hat{s}, s_D \in \{0, 1\}$, we have:

$$p_{\mathcal{O}_{\hat{s}}}(s_D \mid s) = \begin{cases} 1, & \text{if } \left(\sum_{j=1}^{D-1} s_j + s_D\right) \mod 2 = 0 \\ 0, & \text{otherwise} \end{cases}$$

(2)

Suppose first that the algorithm is deterministic, and we choose the prefix $\hat{s}$ uniformly at random. Let us denote by $x_1, x_2, x_3, \ldots, x_q \in \Sigma^*$ the queries to $\mathcal{O}$ generated by the algorithm. The claim is that expected number of queries $q$ needed to ensure at least one $x_i, i \in [q]$ is in $A$ is $2^{D-1}$. Indeed, the $x_i$ s.t. $|x_i| < D - 1$ reveal no information about $\hat{s}$: the output of $\mathcal{O}$ is a uniform Bernoulli random variable regardless of the value of $\hat{s}$. On the other hand, if at some point the algorithm has queried a set $S$ of $x_i$ of length $D-1$, the probability over $\hat{s}$ is uniform over $\Sigma^{D-1} \setminus S$. Hence, the expected number of queries $q$ (expectation being over the choice of $\hat{s}$) a deterministic algorithm needs is lower bounded by $2^{n-1}$.

By Yao's minimax lemma (Yao, 1977), this means that for any (even possibly randomized) algorithm $\mathcal{A}$, there exists $\hat{s}$ on which the algorithm makes at least $2^{n-1}$ queries in expectation. [4] □

Proceeding to the computational lower bound, the theorem we show is as follows:

**Theorem 2.** *There exists a constrained generation task $(\Sigma, A, \mathcal{O})$ for which $\Sigma = \{0, 1\}$, membership in $A \subseteq \Sigma^D$ can be checked in time polynomial in $D$, and $\mathcal{O}$ is such that $\forall s \in \{0, 1\}^D, p_{\mathcal{O}}(s) > 0$, the generation task is NP-hard.*

*Proof.* We construct a reduction from the knapsack problem (Definition 7). Let the set $\{X_1, \ldots, X_D\}$ and the integer $c$ specify an arbitrary instance of the knapsack problem. Consider the constrained generation task specified by $\Sigma := \{0, 1\}$, $A := \{s \in \Sigma^D : \forall i \in [D], s_i \in \{0, 1\}; \sum_{i=1}^D s_i X_i = c\}$. Membership in this $A$ can be clearly verified in polynomial time. Suppose we have a poly-time algorithm that generates a solution $\hat{s}$ to $(\Sigma, A, \mathcal{O})$. Since $\forall s \in \Sigma^D, p_{\mathcal{O}}(s) > 0$, $\hat{s}$ provides a solution to the knapsack problem, as we needed. □

---

[4] We discuss additional intuitions for understanding our proof in Remark 2 in Appendix A.3.

## 4. Constrained generation with process verifier gets easier

While pessimistic, the message of Section 3 agrees with recent developments in inference-time scaling: namely, many natural tasks of interest seem to require a verifier to be solved.

First, we show that the simplest "natural" algorithm with a process verifier, tokenwise rejection sampling (Definition 5), can be much more efficient (exponentially so) in terms of oracle complexity compared to the trivial baseline of rejection sampling (Definition 4).

**Proposition 1.** *Consider the constrained generation task $(\Sigma, A, \mathcal{O})$, s.t. $\Sigma = \{0, 1\}$, $A = \{0^D\}$ and $\mathcal{O}$ is uniform over $\Sigma^D$. Then:*

1. *The expected oracle complexity of rejection sampling (Definition 4) is $2^D D$.*

2. *The expected oracle complexity of tokenwise rejection sampling (Definition 5) with a perfect process verifier is $2D$.*

*Proof.* Both claims are straightforward. (1) follows as generating one guess for the string $s$ takes $D$ oracle calls. Moreover, the probability of the full string matching the only string in $A$ (i.e., $0^D$) is $1/2^D$. As the number of calls to generate $0^D$ is a geometric random variable, the expected number of full string generations is $2^D$.

For (2), since $\mathcal{O}$ is uniform, at each token, the probability of drawing 0 is $1/2$. Hence, the expected number of calls per coordinate needed is 2 — making the total number of expected calls for the entire string $2D$. □

This proposition underscores the power of a process verifier — even in extremely simple settings, and even when used in conjunction with a very simple algorithm.

In fact, one can easily see that with a perfect process verifier, one can easily solve the constrained generation task with $|\Sigma|D$ calls: at each position, one queries the process verifier for each possible continuation of the string, and accepts only if the process verifier accepts. Of course, in practice, the verifier is not perfect, and its accuracy likely depends on how "out-of-distribution" the prefix it's queried on is (See Section 5.1.5 and Appendix B.2.6)

We finally remark that a process verifier, as we defined it, is clearly useful to solve the generation task. If we instead wanted to sample from the restricted distribution $p(s) \propto \mathbf{1}(s \in A)p_{\mathcal{O}}(s)$, it's not clear how useful the process verifier is. For instance, if we use the simple tokenwise rejection sampling (Definition 5), it's easy to see that the distribution we produce samples from is *not* the restricted distribution:

**Proposition 2.** *Consider the constrained generation task* $(\Sigma, A, \mathcal{O})$, *s.t.* $\Sigma = \{0, 1\}$, $A = \{s \in \Sigma^D : \exists i \in [D], s_i = 0\}$ *and* $\mathcal{O}$ *is uniform over* $\Sigma^D$. *Then, tokenwise rejection sampling does* not *produce samples from* $p(s) \propto \mathbf{1}(s \in A)p_{\mathcal{O}}(s)$.

*Proof.* By Definition 5, until the last token is being generated, the process verifier will always accept (as there exists a string with at least one 0 coordinate in the coordinates that haven't yet been sampled). Now, for the prefix $1^{D-1}$, the only completion that is in $A$ is $1^{D-1} \circ 0$. This means that $1^{D-1} \circ 0$ is assigned probability mass $\frac{1}{2^{D-1}}$ under the tokenwise rejection sampling schema. All other strings in $\Sigma^D$ are assigned a probability $\frac{1}{2^D}$. On the other hand, $p(s) \propto \mathbf{1}(s \in A)p_{\mathcal{O}}(s)$ assigns uniform mass on all strings in $A$ — proving the claim of the proposition. $\square$

# 5. Backtracking: a surprisingly effective rejection sampling strategy

The flexibility of the tokenwise rejection sampling with backtracking (Algorithm 1) makes it a very natural strategy to use in conjunction with trained verifiers. We perform a thorough empirical investigation into the applicability of Tokenwise rejection sampling with backtracking in constrained language generation, and benchmark it against common baselines, including rejection sampling (Definition 4), nucleus sampling (Holtzman et al., 2020), temperature scaling, and "block best-of-N" (Appendix B.2.3) sampling, on both synthetic data (Section 5.1) and more realistic data (Section 5.2). We observe that across various settings, Tokenwise rejection sampling with backtracking reduces query complexity, improves accuracy, and does not hurt diversity.

## 5.1. Language models trained on synthetic data

### 5.1.1. DYCK GRAMMAR AS A SANDBOX

Real-world LLM pretraining data (Li et al., 2024a) typically involves many diverse structures, so when an LLM algorithm outperforms baselines on a benchmark, it is generally challenging to precisely identify which component of the algorithm improved the handling of which structures of the data.

To have a quantitative control over the structure in the pretraining data distribution, and to derive fine-grained observations about the effects of Tokenwise rejection sampling with backtracking , we synthetically generate the pretraining data based on the *Dyck grammar* (Schützenberger, 1963), a classic formal language (context-free grammar) consisting of balanced parentheses of multiple types (for example, "[()]" is valid but "([)]" is not). Dyck serves as a useful sandbox, as it typifies features such as long-range dependencies and a hierarchical, tree-like structure—characteristics often found

in both natural and programming language syntax—and has been a subject of interest in numerous theoretical studies on Transformers (Yao et al., 2021; Liu et al., 2022; 2023b; Wen et al., 2023). More formally:

**Definition 8** (Dyck distribution). $\mathsf{Dyck}_D$ *denotes the Dyck language* [5] *of length* $D$ *defined over the vocabulary* $\Sigma = \{\,[\,,\,]\,,\,(\,,\,)\,\}$, *whose* length-$N$ *prefix set is denoted as* $\mathsf{Dyck}_N, \forall N \in [D]$. *For a valid prefix* $w_{1:N} \in \mathsf{Dyck}_N$, *the* depth *of* $w_{1:N}$ *is*

$$d(w_{1:N}) = \#\textit{Open Brackets in } w_{1:N}$$
$$- \#\textit{Close Brackets in } w_{1:N}.$$

*The distribution* $\mathcal{D}_{\mathsf{Dyck}}$ *over* $\mathsf{Dyck}_N$, *(parameterized by* $p, q \in (0, 1)$*) is defined such that* $\forall w_{1:N} \in \mathsf{Dyck}_N$,

$$\mathbb{P}(w_{1:N}) \propto p^{|\{i|w_i=[,d(w_{1:i})=1\}|} \cdot (1-p)^{|\{i|w_i=(,d(w_{1:i})=1\}|} \tag{3}$$

$$\cdot (pq)^{|\{i|w_i=[,d(w_{1:i})>1\}|} \cdot ((1-p)q)^{|\{i|w_i=(,d(w_{1:i})>1\}|} \tag{4}$$

$$\cdot (1-q)^{|\{i|w_i\in\{],)\},d(w_{1:i})\leq D-i\}|}.$$

**Remark 1.** *Equation* (3) *defines an intuitive autoregressive generative process for* $\mathsf{Dyck}_D$*: if the current depth is 0, then sample the next token from* $[$ *and* $($ *with probability* $p$ *and* $1 - p$ *respectively; else if the current depth is* $D - i + 1$, *implying that all the remaining positions have to be closing brackets, then deterministically close the last unmatched open bracket* [6]*; else, sample the next token from open or close brackets with probability* $q$ *and* $1 - q$ *respectively. In other words,* $p$ *controls the proportion of square vs. round brackets, while* $q$ *controls the tendency to predict an open bracket when possible (a large* $q$ *may result in a large depth at some position).*

In our experiments, we pretrain autoregressive Transformer (Vaswani et al., 2017) Language models (6 layers, 8 heads per layer, hidden dimension 512) from scratch on data sampled from $\mathcal{D}_{\mathsf{Dyck}}$ with $D = 32, p = 0.2, q = 0.5$. We use batch size 32, weight decay 0.1, learning rate 3e-4 with 100 warmup steps, and follow Block et al. (2024) to use exponential moving average to stabilize training. We reached 100% training and (in-distribution) validation accuracy.

To search for stronger signals in benchmarking the accuracy of the trained model, we will prompt it using the following type of *out-of-distribution* prompts. Note that since $p < 0.5$, the training data contains less square brackets than round brackets, so long prefixes with many square brackets will be *out-of-distribution* prompts for the trained model. We generated a set of such out-of-distribution prompts $\mathsf{Dyck}_{OOD}$

---

[5]We follow a simplified version of Wen et al. (2023) in defining a probability distribution over strings in a Dyck language.

[6] At any position, there is at most one valid closing bracket.

from $\mathsf{Dyck}_N$ with $p = 0.8$ where the prefix length $N$ is uniformly randomly sampled from $25 \leq N \leq 31$. We let the trained language model complete these prompts and check whether the completed string is in $\mathsf{Dyck}_D$. Quantitatively:

**Definition 9** (Prompt completion accuracy). *Given an autoregressive oracle $\mathcal{O}$ (Definition 1) and a set of prefix prompts $X$, the accuracy of $\mathcal{O}$ in completing $X$ is:*

$$Acc(\mathcal{O}, X) = \frac{1}{|X|} \sum_{x \in X, y \sim p_{\mathcal{O}}(\cdot|x)} \mathbf{1}_{x \circ y \in \mathsf{Dyck}_D}$$

We construct the autoregressive oracle $\mathcal{O}_{\text{nucleus}}$ which predicts the next-token distribution based on our trained model with nucleus sampling (Holtzman et al., 2020) top_p set to 0.9. We observed that $Acc(\mathcal{O}_{\text{nucleus}}, \mathsf{Dyck}_{OOD}) = 94.23\%$. We will show that $\mathcal{O}_{\text{verifier backtracking}}$ based on Algorithm 1 can significantly reduce the remaining error rate.

### 5.1.2. TRAINING THE VERIFIER

We collect a set of 441 prompts in $\mathsf{Dyck}_{OOD}$ in which the trained model (denoted as LM) made mistakes when completing them. We implement a rule-based error parser according to the grammars of $\mathsf{Dyck}_D$ which identifies the first position of error in each model completion. Applying this parser to the model mistakes, we obtain a set of model-generated strings $X_{\text{error}} \subset \Sigma^*$ which contain errors. By contrast, we sample another set of 441 strings $X_{\text{correct}} \sim \mathsf{Dyck}_{OOD}$ such that $X_{\text{error}}$ and $X_{\text{correct}}$ have the same length distribution. We train a lightweight neural network verifier to distinguish $X_{\text{error}}$ from $X_{\text{correct}}$.

Concretely, to maximally exploit the representations learned by LM, we train a 1-linear-layer verifier V whose features are the last-layer-last-position representations by LM of strings in $X_{\text{error}} \cup X_{\text{correct}}$, and labels are 0 for strings in $X_{\text{error}}$ and 1 for strings in $X_{\text{correct}}$. Consequently, the trainable parameters of V are a single matrix of dimensionality 512 by 2. Among the 882 strings in $X_{\text{error}} \cup X_{\text{correct}}$, we use 792 samples for training, and 90 samples for validation. Despite being slightly over-parameterized, this minimal verifier V achieved on average 93% (with standard error 3.9%) validation accuracy across 10 repetitions. Figure 1 in Appendix B.1.1 illustrates the intuition of why a lightweight verifier may be surprisingly effective with a small number of labeled samples. We next verify that forcing a backtracking at prefixes where the model made mistakes can effectively improve the completion accuracy (Section 5.1.3), and that the trained verifier in this section can mostly catch those mistakes and thus mostly retaining the accuracy gain (Section 5.1.4).

### 5.1.3. BACKTRACKING EFFECTIVELY REDUCES ERRORS

The trained language model LM made a mistake at the last position of each string $x \in X_{\text{error}}$. We therefore use "error-

inducing prefixes" $X_{\text{error-inducing}}$ to denote $\{x_{1:|x|-1} \mid x \in X_{\text{error}}\}$. Table 1 shows that at prefixes in $X_{\text{error-inducing}}$, if we backtrack *only once* for a small backtrack stride $B$, and continue the autoregressive sampling process, the error rate can be significantly reduced.

| generation configuration | accuracy |
|---|---|
| baseline: nucleus sampling top_p = 0.9 | 0.331 |
| baseline: greedy argmax sampling | 0.334 |
| $B = 1$, then nucleus sampling top_p = 0.9 | 0.366 |
| $B = 2$, then nucleus sampling top_p = 0.9 | 0.438 |
| $B = 4$, then nucleus sampling top_p = 0.9 | 0.591 |
| $B = 8$, then nucleus sampling top_p = 0.9 | 0.790 |

*Table 1.* At error-inducing prefixes, a larger backtrack stride $B$ significantly improves completion accuracy (Definition 9).

### 5.1.4. VERIFIER EFFECTIVELY REDUCES ERRORS

In Section 5.1.3, the sampling process forced a backtracking at error-inducing prefixes $X_{\text{error-inducing}}$. Can the error reduction effect be retained by a *trained* lightweight single-layer verifier V in Section 5.1.2? Table 2 shows that Tokenwise rejection sampling with backtracking (Algorithm 1) using the trained verifier is remarkably effective. Moreover, in Appendix B.1.2, we verify that the predicted backtracks were necessary.

| $Q$ | $B$ | accuracy |
|---|---|---|
| 1 | 2 | 0.421 |
|  | 4 | 0.500 |
|  | 6 | 0.604 |
| 2 | 2 | 0.457 |
|  | 4 | 0.634 |
|  | 6 | 0.762 |
| 4 | 2 | 0.518 |
|  | 4 | 0.762 |
|  | 6 | 0.921 |
| baseline: nucleus sampling top_p = 0.9 |  | 0.331 |
| baseline: greedy argmax sampling |  | 0.334 |

sampling

*Table 2.* When the prompts are error-inducing prefixes, a single-layer trained verifier significantly improves completion accuracy using Tokenwise rejection sampling with backtracking (Algorithm 1). A larger backtrack quota $Q$ and a larger backtrack stride $B$ are both helpful.

### 5.1.5. TOKENWISE REJECTION SAMPLING WITH BACKTRACKING REDUCES COMPLETION ERRORS ON UNSEEN OOD PREFIXES

Table 2 in Section 5.1.4 reported a significant improvement of accuracy by Tokenwise rejection sampling with backtracking (Algorithm 1) when the prompts are $X_{\text{error-inducing}}$, for which the language model LM made mistakes during

completion. Is the verifier V overfitted to these type of error-inducing prompts? Can the accuracy improvement generalize to (average-case) out-of-distribution (OOD) prefixes, i.e. independently sampled strings of the same distribution as $\text{Dyck}_{OOD}$ (Section 5.1.1)?

We independently sampled 10000 such out-of-distribution prompts $\text{Dyck}_{OOD}^{unseen}$, and benchmark the accuracy of Tokenwise rejection sampling with backtracking (Algorithm 1) against the baselines of nucleus sampling top_p = 0.9 (Holtzman et al., 2020) and standard autoregressive sampling (equivalent to top_p = 1.0). Table 3 shows that Tokenwise rejection sampling with backtracking (Algorithm 1) significantly reduces completion errors. Crucially, the improvement does not diminish on top of the commonly used baseline of truncating the tail probabilities during sequence sampling. This verifies the desirable property that Tokenwise rejection sampling with backtracking can be applied in combination with such baselines to further improve accuracy. We also verify that the accuracy improvement does not hurt diversity (Appendix B.1.4).

Finally, provided with the verifier, why does the model still make mistakes? We include additional error analysis in Appendix B.1.3.

| nucleus sampling top_p | $Q$ | $B$ | #errors $\pm$ std err |
|---|---|---|---|
| 0.9 | 0 | 0 | $240.0 \pm 5.177$ |
| | 4 | 4 | $179.4 \pm 1.020$ |
| 1.0 | 0 | 0 | $461.8 \pm 8.304$ |
| | 4 | 4 | $200.0 \pm 3.225$ |

*Table 3.* Tokenwise rejection sampling with backtracking (Algorithm 1) reduces completion errors on unseen out-of-distribution (OOD) prefixes. Crucially, the improvement does not diminish on top of commonly used baselines, including nucleus sampling top_p = 0.9. For each setting of top_p, we compare Tokenwise rejection sampling with backtracking (Algorithm 1) (using backtrack quota $Q = 4$ and backtrack stride $B = 4$) with the baseline (using backtrack quota $Q = 0$ and backtrack stride $B = 0$). We report the number of completion errors that occur when completing an unseen set of 10000 independently sampled out-of-distribution prompts $\text{Dyck}_{OOD}^{unseen}$. The experiment was repeated 5 times, and we report the standard errors.

**5.2. Generating test cases with pretrained CodeLlama**

Motivated by our findings in Section 5.1, we apply essentially the same recipe of Tokenwise rejection sampling with backtracking (Algorithm 1) to a real-data use case, and show that Algorithm 1 clearly improves the quality vs. query complexity trade-off on top of commonly used baselines, such as nucleus sampling (Holtzman et al., 2020), temperature scaling, best-of-n rejection sampling, and block best-of-n with process reward model.

5.2.1. TASK SETUP

A natural practical constrained generation task that requires both accuracy and diversity is generating test cases for a target function specified by the prompt. To have an unambiguous notion of groundtruth regarding accuracy and diversity, we control the target function to be a simple implementation of the `append` function for Python lists. Under this setting, we wrote an evaluator script which analyzes model generated completions, measuring the accuracy by checking whether a test case correctly tests list `append`, and measuring the diversity by checking how many distinct test cases are generated. [7]

We write a program to systematically generate task prompts, randomizing over function names and demonstration examples. Each prompt includes 1 demonstration example specifying the intended output format, followed by a target function (implementing `append`), and finally requests 8 test cases be generated. Two examples of the prompt are provided in Table 7, and correspondingly, two examples of model completions of these prompts are provided in Table 8 in Appendix B.2.1.

**Evaluation metrics** The test prompts include 10 different target function names that are unseen during training. Each target function name is independently tested 10 times. Since each prompt requests 8 test cases, the total number of test cases requested for each run of a decoding algorithm is $8 \times 10 \times 10 = 800$. We will measure the following metrics:

1. $N_{\text{distinct correct}}$: the number of **distinct correct** test cases generated. This metric naturally incorporates both accuracy and diversity.

2. $\text{Acc}_{\text{distinct}} \coloneqq N_{\text{distinct correct}}/800$.

3. $\mathcal{C}$: the query complexity (analogous to Definition 6). We measure the total number of queries made to the generator LM when it completes the prompts. Each completion allows at most 384 tokens to be generated, so the max $\mathcal{C}$ is $384 \times 10 \times 10 = 38400$ unless "block best-of-n" (Appendix B.2.3) is used.

We use a pretrained CodeLlama (Roziere et al., 2023) as the generator language model LM, which we freeze during our experiments. We discuss common baselines in Appendix B.2.2. We follow almost the same approach as Section 5.1.2 to train our verifier on this coding task. We next present technical details and ablation experiments regarding design choices of verifier training in Appendix B.2.3.

---

[7]Two test cases are different if and only if they test different lists or different appended items.

| $Q$ | $B$ | top_p | T | block BoN | $\text{Acc}_{\text{distinct}} \pm$ std err |
|---|---|---|---|---|---|
| 4 | 4 | 0.95 | 1.0 | | $0.714 \pm 0.011$ |
| 0 | | 0.95 | 1.0 | 2 | $0.684 \pm 0.038$ |
| 0 | | 0.95 | 1.0 | | $0.660 \pm 0.042$ |
| 0 | | 0.95 | 1.0 | 4 | $0.623 \pm 0.036$ |
| 0 | | 0.95 | 1.0 | 8 | $0.559 \pm 0.038$ |
| 4 | 4 | 1.0 | 1.0 | | $0.639 \pm 0.061$ |
| 4 | 10 | 1.0 | 1.0 | | $0.622 \pm 0.046$ |
| 0 | | 1.0 | 1.0 | | $0.504 \pm 0.025$ |
| 4 | 4 | 1.0 | 1.2 | | $0.440 \pm 0.026$ |
| 0 | | 1.0 | 1.2 | | $0.269 \pm 0.025$ |
| 0 | | 0.0 | 1.0 | | $0.013 \pm 0.000$ |

*Table 4.* Tokenwise rejection sampling with backtracking (Algorithm 1) improves accuracy and outperforms nucleus sampling top_p, temperature scaling T, and block best-of-n (BoN) (Appendix B.2.3). In this table, we divide the rows into groups, separated by double horizontal lines, such that each group uses the same top_p and temperature. The backtrack quota $Q = 0$ means a baseline algorithm that does not use the verifier. $Q > 0$ means Tokenwise rejection sampling with backtracking with the corresponding $Q$ and $B$. *block BoN* specifies the number of candidates generated for each block; empty block BoN means not using block best-of-n. In all groups, Tokenwise rejection sampling with backtracking leads to higher $\text{Acc}_{\text{distinct}}$ than all other methods. The last group corresponds to argmax greedy decoding, which has low $\text{Acc}_{\text{distinct}}$ due to low diversity. The experiment was repeated 5 times, and we report the standard errors. The complete set of experiments are reported in a larger Table 13 in Appendix B.2.4.

### 5.2.2. TOKENWISE REJECTION SAMPLING WITH BACKTRACKING IMPROVES ACCURACY

In this section, we show that Tokenwise rejection sampling with backtracking (Algorithm 1) achieves higher $\text{Acc}_{\text{distinct}}$ than all the baselines described in Appendix B.2.2. Similar to our observations based on the synthetic Dyck grammar data (Section 5.1.5), the improvement does not diminish on top of commonly used baselines. This verifies the desirable property that Tokenwise rejection sampling with backtracking (Algorithm 1) can be applied in combination with commonly used baselines to further improve accuracy. The primary comparisons are reported in Table 4, and additional results are in Table 13 in Appendix B.2.4. Moreover, in Appendix B.2.6, we show that analogous to our observations on the synthetic Dyck grammar (Section 5.1.5), Tokenwise rejection sampling with backtracking (Algorithm 1) generalizes better to *out-of-distribution* prompts than baselines.

### 5.2.3. TOKENWISE REJECTION SAMPLING WITH BACKTRACKING IS QUERY EFFICIENT

In this section, we show that Tokenwise rejection sampling with backtracking (Algorithm 1) achieves a better tradeoff between $\text{Acc}_{\text{distinct}}$ and query efficiency $\mathcal{C}$ than all the baselines described in Appendix B.2.2. The primary com-

parisons are visualized in Figure 3 and Figure 4 (in Appendix B.2.5). Numerical values of $\mathcal{C}$ are reported in Table 13 in Appendix B.2.4.

## 6. Related work

**Inference-time scaling for language models** Practical language generation tasks typically impose various task-specific constraints in addition to the general grammatical rules of language. One effective way to improve the chance of satisfying such constraints is to increase the inference-time compute through search and/or rejection sampling. There has been a long history of prior works that employ inference-time scaling in the language generation context, dating as far back as beam search (Lowerre & Reddy, 1976; Hayes-Roth et al., 1976; Ow & Morton, 1988; Jurafsky & Martin, 2000; Graves, 2012). Much more recently, as researchers develop the techniques for language models to follow instructions (see the survey by Zhang et al. (2023b) and references therein), more creative designs for inference-time scaling algorithms have become viable (Wang et al., 2022; Yao et al., 2023; Zhang et al., 2023a; Zhou et al., 2023; Choi et al., 2023; Liu et al., 2024; Xie et al., 2024; Snell et al., 2024; Zhao et al., 2024), and see Wu et al. (2024) for a recent survey on cost-performance tradeoffs of these approaches. Compared to these approaches in the literature, our Tokenwise rejection sampling with backtracking (Algorithm 1) shares some features with lookahead search (Snell et al., 2024) (specifically, the rejection decision at the current position is based on the verifier decision at some future position). However, two main differences are: (1) Tokenwise rejection sampling with backtracking (Algorithm 1) does not use a beam (i.e. does not need to generate multiple candidates, thus reducing query complexity), and (2) Algorithm 1 uses a different sampling approach (namely argmax) for the backtracked positions (we verify in Appendix B.3 that in some settings this significantly improves the accuracy). It is a natural future research direction to design inference algorithms that combine the advantages of the two.

**Incorporating a process reward model to assist language generation** Among the vast design space for inference-time scaling, process reward modeling has been proven to be an important component common to many LLM systems (Polu & Sutskever, 2020; Uesato et al., 2022; Ma et al., 2023; Lightman et al., 2023; Wang et al., 2024). The process verifier which we study (Definition 3) is a special case of such process reward model if we restrict the output to be binary. However, there are still challenging open problems around process reward modeling, such as how to properly define the "blocks" (Guo et al., 2025) (see also our definitions in the "Block verifier" part of Appendix B.2.3). Towards bringing more clarity to these open questions, our

work develops a theoretical framework for reasoning about the query complexity of process verifiers. Moreover, our experiments suggest the potentials of a lightweight process verifier in improving the query complexity, accuracy, and diversity of constrained generation. In particular, our theory and experiments suggest (1) the "blocks" do not necessarily have to be carefully designed — setting each token as a block might potentially suffice, at least in some more structured domains such as codes; (2) *backtracking* (Algorithm 1, Section 5) is a robustly effective strategy that should be applied in combination with process verifiers. A possible extension of the type of verifier we study (Definition 3) is: instead of outputting binary acceptance / rejection decisions, the verifier could return a probability of accepting each prefix (Yang & Klein, 2021). However, some tasks may require that the output distribution should match some target distribution, and it may be challenging to ensure that the acceptance probability is well-calibrated in order to satisfy this requirement.

**Controlled synthetic data distribution as a sandbox for studying language models**  Our Dyck grammar distribution most closely follows Wen et al. (2023) (though we switched to a fixed-sequence-length setting, and used unbalanced bracket type probability, instead of length extrapolation, to define the criteria for a prompt to be *out-of-distribution*). Dyck grammar was also used in other prior works (Hewitt et al., 2020; Ebrahimi et al., 2020; Yao et al., 2021; Liu et al., 2022; 2023b) to study language models. Dyck grammar can be seen as a special case of the task (specifically context-free grammar) considered in SynCode (Ugare et al., 2024). Other synthetic data distributions have been used to study various aspects of language models in prior works, including representational capability (Bhattamishra et al., 2020; Li & Risteski, 2021; Zhang et al., 2022; Zhao et al., 2023), statistical sample complexity (Edelman et al., 2022), optimization process (Lu et al., 2021; Jelassi et al., 2022; Li et al., 2023; Bietti et al., 2023), sampling (Li et al., 2024b), and architectural limitations (Liu et al., 2023a), and see references cited therein.

## 7. Conclusion

We introduce a new theoretical framework for elucidating the design space of verifiers and correspondingly a simple family of rejection-sampling-based inference algorithms. In particular, our theory proves the computational benefits of incorporating a *process verifier*, measured by the *query complexity* of calling the generator. On the other hand, our theory also reveals the subtleties: straightforwardly applying a process verifier in a Tokenwise rejection sampling algorithm may unintentionally re-weigh the distribution among sequences that satisfy the constraints, which could be undesirable for settings that require a strong notion of dis-

tributional *calibration*. Empirically, through fine-grained experiments on both synthetic and realistic data, we show that the Tokenwise rejection sampling algorithm, when combined with *backtracking*, is a robustly effective recipe for reducing query complexity, improving accuracy, and maintaining diversity. For future works, we hope the theoretical framework and empirical observations can inspire systematic characterization of the strengths and weaknesses of the diverse set of rejection-sampling-based inference-time algorithms. Concrete open problems at the intersection of theory and experiments include investigating the realistic and necessary conditions on the verifiers for the inference-time algorithm to achieve distributional calibration (e.g. it is unrealistic in some language generation setting to assume that a verifier returns the calibrated acceptance probability in rejection sampling), and synergistically designing query-efficient verifier-assisted generation algorithms.

## ACKNOWLEDGEMENTS

We thank Bingbin Liu for insightful discussions.

AR and YL are supported in part by NSF awards IIS-2211907, CCF-2238523, IIS-2403275, an Amazon Research Award, a Google Research Scholar Award, an OpenAI Superalignment Fast Grant, and DoD Award N000142512124.

Part of this work was done when YL and AR were visiting the Simons Institute for the Theory of Computing. We thank the Simons Institute for hosting us. We also thank the Simons Foundation and Google for sponsoring computational resources for us and other program visitors, and we thank Matus Telgarsky and NYU IT for managing these computational resources.

## Impact Statement

This paper presents work whose goal is to advance the field of Machine Learning. There are many potential societal consequences of our work, none of which we feel must be specifically highlighted here.

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

# Supplementary Material

## A. Discussions

### A.1. Is query efficiency a reasonable notion of efficiency?

There are many reasonable efficiency metrics, and they do not always positively correlate with each other (Dehghani et al., 2021).

Our paper focuses on *query complexity* (measured by the number of tokens generated by the language model to satisfactorily complete the task [8] ), and we do not claim that the same conclusions apply when we switch out query complexity for other metrics of efficiency, such as wall-clock time.

We think query complexity is one (but not necessarily the only, or the most) important aspect of efficiency due to the following considerations:

- Many existing large language model (LLM) providers charge service fees to the users according to the number of tokens generated by the language model for the user, i.e. query complexity.

- In the *single sequence generation* setting, controlling all other conditions to be held the same, query complexity positively correlates with the size of computation (the number of decoder forward passes) and wall-clock time.

- In the *batched generation* setting, admittedly, the wall-clock time does not necessarily scale linearly with query complexity [9] , meaning that the naive best-of-$n$ rejection sampling is not as slow as query complexity would indicate (if the LLM has sufficient bandwidth for it). However, in many realistic LLM inference settings, the LLM receives a large number of query requests per second, so there is no additional idle availability [10] for duplicating each sequence generation request by $n$.

Although, as mentioned above, query complexity is partially indicative of a few practically important efficiency metrics (e.g. monetary cost or wall-clock time), there are aspects of these metrics that are not tracked by query complexity. For example, different types of *hardware* and *cache* may have different efficiency best practices. In particular, on GPUs and TPUs, algorithms that better exploit *parallelization* or *tensorized computation* tend to be more efficient. Therefore, an important direction for future work is to design and analyze *hardware-aware algorithms* that incorporate these important aspects of the inference setup.

---

[8]This definition is natural since generating one token involves one forward pass of the (decoder-only autoregressive) language model, i.e. one query.

[9]For example, the wall-clock time of generating $n$ candidate responses (with batch size $n$) might be less than $n$ multiplying the wall-clock time of generating 1 candidate response.

[10]Unless more GPUs/TPUs are allocated to serve this LLM.

## A.2. On the hardness of the knapsack problem

The hardness of the knapsack problem has been the subject of extensive study. Specifically, the decision version of this problem has found applications in the context of secure cryptosystems (Odlyzko, 1998). Under no assumptions on the input structure, the best known algorithm is based on dynamic programming (Kellerer et al., 2004) and runs in pseudo-polynomial time. This algorithm is also used to obtain an FPTAS and its runtime is effectively polynomial if one further assumes that the weights are polynomially bounded in $D$. More exact or approximate algorithms achieve polynomial runtime, under specific input structures. Specifically, when the weights form a superincreasing sequence, that is,

$$X_i \geq \sum_{j=1}^{i-1} X_j \ \ \forall i \in [2, D] \cap \mathbb{Z},$$

a greedy algorithm solves the knapsack decision problem (Odlyzko, 1998) in linear time. On the other hand, when the density of the knapsack

$$\frac{D}{\log_2(\max_i\{X_i\}_{i=1}^d)}$$

is small enough, knapsack is approximately solved in polynomial time by lattice reduction algorithms (Plantard et al., 2013). Our argument considers the most general setting, in which no assumptions are made on the structure of the inputs $\{X_i\}_{i=1}^t$, $c$ and the decision problem is NP-complete (Karp, 1972).

## A.3. Proof intuition for Theorem 1

**Remark 2.** *To help readers parse our proof above, we provide its informal intuition. The oracle $\mathcal{O}_{\hat{s}}$ can be thought of as hiding a secret message $\hat{s}$ which is a binary sequence of length $D - 1$. Because of our construction in Equation (1), by querying the oracle with any string $s \neq \hat{s}$, the output of the oracle will not reveal any information about $\hat{s}$. Therefore, to know anything about $\hat{s}$, the query $s$ needs to exactly match $\hat{s}$. For any deterministic order of searching for $\hat{s}$ over $\{0, 1\}^{D-1}$, the worst case is always that $\hat{s}$ is the last item in the search order, causing runtime $2^{D-1}$.*

*Moreover, Theorem 1 generally applies to any algorithm $\mathcal{A}$. In particular, $\mathcal{A}$ is even allowed to* never *query the generator oracle at all. Intuitively, one candidate counter-example of our theory would be a simple algorithm $\mathcal{A}$ which always outputs 0 at all positions (as this will obviously satisfy the counstraint $A := \{s \in \Sigma^D : \sum_{i=1}^{D} s_i \mod 2 = 0\}$). However, recall that Definition 2 requires that the output must have nonzero probability under the generator oracle $\mathcal{O}_{\hat{s}}$. Note that $s_i = 0 \ \forall i \in [D]$ will not have nonzero probability under $\mathcal{O}_{\hat{s}}$, thus violating the constraints (unless $\hat{s}_i = 0 \ \forall i \in [D - 1]$ ). The intuitive reason why the above counter-example does not work is that, it is necessary to use the oracle $\mathcal{O}_{\hat{s}}$ to know the $\hat{s}$ that it hides. Therefore, any algorithm $\mathcal{A}$ is governed by the above-mentioned lower bound of searching for $\hat{s}$ by querying $\mathcal{O}_{\hat{s}}$.*

# B. Additional experimental results

We complement Section 5 by providing additional technical details.

## B.1. Additional results about language models trained on synthetic data

### B.1.1. VISUALIZING THE LANGUAGE MODEL REPRESENTATIONS OF CORRECT VS. INCORRECT SEQUENCES

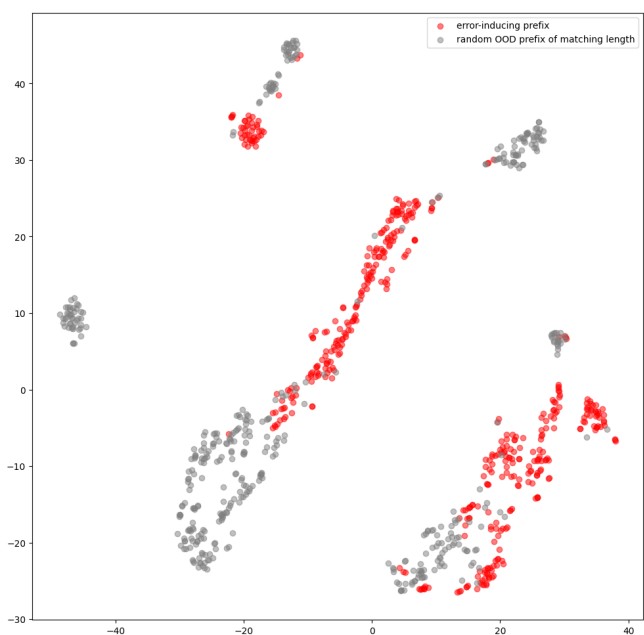

*Figure 1.* TSNE plot for the LM last-layer-last-position representations of strings in $X_{\text{error}} \cup X_{\text{correct}}$. Red dots correspond to the representations of incorrect strings, whereas gray dots correspond to the representations of correct strings of comparable lengths. We can see that the representations of incorrect strings form just a few clusters. This intuitively justifies using a lightweight verifier on top of these LM representations.

B.1.2. THE PREDICTED BACKTRACKS WERE NECESSARY

During the experiment in Section 5.1.4, the trained verifier V predicted backtracks at many positions. Were they really necessary? For each setting of backtrack quota $Q$ and backtrack stride $B$, we collect the set of prefixes $X_{\text{predicted backtracks}}$ where V predicted backtracks. Then, we let the language model LM complete each string in $X_{\text{predicted backtracks}}$ without any backtracks, using common decoding techniques such as nucleus sampling top_p = 0.9 (Holtzman et al., 2020) and argmax greedy decoding. Table 5 shows that without backtracking, the completion accuracy is much lower than the accuracy reported in Table 2. This implies that $X_{\text{predicted backtracks}}$ were indeed challenging prefixes for the LM, which verifies that the backtracks predicted by verifier V were necessary.

| $Q$ | $B$ | #backtracks | accuracy without backtrack | |
|---|---|---|---|---|
| | | | nucleus sampling top_p = 0.9 | argmax |
| 1 | 2 | 163 | 0.313 | 0.344 |
| | 4 | 163 | 0.337 | 0.319 |
| | 6 | 163 | 0.331 | 0.288 |
| 2 | 2 | 311 | 0.347 | 0.328 |
| | 4 | 297 | 0.357 | 0.349 |
| | 6 | 286 | 0.374 | 0.373 |
| 4 | 2 | 600 | 0.371 | 0.353 |
| | 4 | 532 | 0.419 | 0.404 |
| | 6 | 489 | 0.509 | 0.523 |

*Table 5.* Predicted backtracks were necessary. For each setting of backtrack quota $Q$ and backtrack stride $B$, we report the number of times that Tokenwise rejection sampling with backtracking (Algorithm 1) backtracked. Moreover, we report the completion accuracy of letting the language model LM complete these backtracked prefixes without any backtrack. For each setting, the completion accuracy is much lower than the accuracy reported in Table 2. This implies that these backtracked prefixes were indeed challenging prefixes for the LM.

B.1.3. Error analysis on the remaining mistakes

Given the improvement of accuracy (Section 5.1.5) as a result of our algorithm Tokenwise rejection sampling with backtracking (Algorithm 1), why did the model still make mistakes?

We conducted an error analysis which parses all mistakes into error types, and examine the generated token, the LM predicted most probable token, their predicted probabilities, and a few intermediate variables during the course of our algorithm Tokenwise rejection sampling with backtracking (Algorithm 1).

In summary, the findings are:

1. Among 225 generated mistakes, 222 correspond to predicting an incorrect closing bracket, and 3 correspond to pre-maturely predicting the end-of-sequence <eos> token.

2. In all 225 cases, the final state of the algorithm has used up all the backtrack quota $Q$ allocated to it, so even if the error predictor was perfect, the algorithm would not have been had a chance to correct these mistakes. This suggests that suitably increasing backtrack quota $Q$ might be an effective approach in improving the accuracy (though there are trade-offs with query efficiency).

A snapshot of our error analysis result is included in Figure 2. We have released our experimental codes, [11] which include more error analyses.

| | error | prefix | generated_token | generated_token_prob | most_probable_token | most_probable_token_prob | backtrack_quota |
|---|---|---|---|---|---|---|---|
| 0 | INCORRECT_CLOSING_BRACKET | B(((((([([()))](()))((]()) | ] | 0.998425 | ] | 0.998425 | 0 |
| 1 | INCORRECT_CLOSING_BRACKET | B([()](((()))((()()))()))( | ] | 0.901743 | ] | 0.901743 | 0 |
| 2 | INCORRECT_CLOSING_BRACKET | B(()((0[])())0(((())()()( | ] | 0.684750 | ] | 0.684750 | 0 |
| 3 | INCORRECT_CLOSING_BRACKET | B(((())0(())(0(()()((())) | ] | 0.699475 | ] | 0.699475 | 0 |
| 4 | INCORRECT_CLOSING_BRACKET | B(([((())0()))](())](()()( | ] | 0.994803 | ] | 0.994803 | 0 |
| 5 | INCORRECT_CLOSING_BRACKET | B(()(()((()((0()())()))()( | ] | 0.987031 | ] | 0.987031 | 0 |
| 6 | INCORRECT_CLOSING_BRACKET | B(()[(((()))())]]((())( | ] | 0.869623 | ] | 0.869623 | 0 |
| 7 | INCORRECT_CLOSING_BRACKET | B(()(((0000)(()))]0(() | ] | 0.782469 | ] | 0.782469 | 0 |
| 8 | INCORRECT_CLOSING_BRACKET | B](((((([[()))0(()))))0[(0 | ] | 0.802167 | ] | 0.802167 | 0 |
| 9 | INCORRECT_CLOSING_BRACKET | B((((([])))))0()0([[() | ] | 0.941579 | ] | 0.941579 | 0 |
| 10 | INCORRECT_CLOSING_BRACKET | B(()(((((([])))))0()(() | ] | 0.997442 | ] | 0.997442 | 0 |
| 11 | INCORRECT_CLOSING_BRACKET | B(((())000(0((((())()))) | ] | 0.965299 | ] | 0.965299 | 0 |
| 12 | INCORRECT_CLOSING_BRACKET | B[((()))(0(())(()())])(() | ] | 0.523672 | ] | 0.523672 | 0 |
| 13 | INCORRECT_CLOSING_BRACKET | B(()([[]]0000[(())(() | ] | 0.638692 | ] | 0.638692 | 0 |
| 14 | INCORRECT_CLOSING_BRACKET | B(()()(((()))((())()(() | ] | 0.995885 | ] | 0.995885 | 0 |
| 15 | INCORRECT_CLOSING_BRACKET | B([]0(((())(()000[]))(() | ] | 0.928873 | ] | 0.928873 | 0 |
| 16 | INCORRECT_CLOSING_BRACKET | B(((())((((((((()0())))))))) | ] | 0.802617 | ] | 0.802617 | 0 |
| 17 | INCORRECT_CLOSING_BRACKET | B(())((()(())[[()])((()) | ] | 0.864548 | ] | 0.864548 | 0 |
| 18 | INCORRECT_CLOSING_BRACKET | B((()))(])(((())(()(() | ] | 0.722893 | ] | 0.722893 | 0 |
| 19 | INCORRECT_CLOSING_BRACKET | B(()(()))(()((()(()))) | ] | 0.963540 | ] | 0.963540 | 0 |
| 20 | INCORRECT_CLOSING_BRACKET | B(())(((((())))(()()()( | ] | 0.932594 | ] | 0.932594 | 0 |
| 21 | END_INCORRECT | B(((())(())()(()())(() | E | 0.975265 | E | 0.975265 | 0 |
| 22 | INCORRECT_CLOSING_BRACKET | B((()[])))(()(())))0(()) | ] | 0.975087 | ] | 0.975087 | 0 |

*Figure 2.* Error analysis table for mistakes of language model trained on Dyck grammar and sampled using Tokenwise rejection sampling with backtracking (Algorithm 1). The last column records the remaining backtrack quota $Q$ at the time of generating the incorrect token.

---

[11] https://github.com/YuchenLi01/LM_Query_Complexity

B.1.4. TOKENWISE REJECTION SAMPLING WITH BACKTRACKING MAINTAINS DIVERSITY

In this section, we show that the significant accuracy improvement is not at the cost of reducing diversity.

Our experiment freshly samples 100 prompts following the same distribution as $\text{Dyck}_{\text{OOD}}$ (Section 5.1.1). For each prompt, we let the trained LM independently sample 10 completions, using Tokenwise rejection sampling with backtracking (Algorithm 1) or the baseline algorithm, and will compare how many (out of 10) samples were different, and report the mean and standard error across the 100 prompts.

Table 6 shows that Tokenwise rejection sampling with backtracking (Algorithm 1) generates similarly diverse samples as the baselines of nucleus sampling with top_p = 0.9 or 1.0.

| $Q$ | $B$ | top_p | diversity $\pm$ std err (out of 10) |
|---|---|---|---|
| 4 | 4 | 1.0 | $5.52 \pm 3.28$ |
| 0 | 0 | 0.9 | $5.47 \pm 3.06$ |
| 0 | 0 | 1.0 | $5.84 \pm 3.29$ |

*Table 6.* Under the experiment setup described in Appendix B.1.4, Tokenwise rejection sampling with backtracking (Algorithm 1) is similarly diverse as the baselines of nucleus sampling with top_p = 0.9 or 1.0.

## B.2. Additional results about generating test cases with pretrained CodeLlama

This section complements our results in Section 5.2.

### B.2.1. EXAMPLES OF PROMPTS AND MODEL COMPLETIONS

---

```
def f(a, b):
        return a + b

List 8 test cases of the above function f, one in each line:
assert f(5, 5) == 10
assert f(1, 5) == 6
assert f(2, 8) == 10
assert f(6, 2) == 8
assert f(6, 9) == 15
assert f(4, 5) == 9
assert f(9, 6) == 15
assert f(6, 1) == 7

def knk(l, item):
        assert type(l) is list
        l.append(item)
        return l

List 8 test cases of the above function knk, one in each line:
```

---

```
def f(a, b):
        return a + b

List 8 test cases of the above function f, one in each line:
assert f(5, 8) == 13
assert f(1, 5) == 6
assert f(8, 4) == 12
assert f(6, 2) == 8
assert f(3, 9) == 12
assert f(1, 7) == 8
assert f(5, 9) == 14
assert f(1, 7) == 8

def ovs(l, item):
        assert type(l) is list
        l.append(item)
        return l

List 8 test cases of the above function ovs, one in each line:
```

---

*Table 7.* Two example prompts for generating test cases for a simple implementation of the `append` function for Python lists. In these examples, the following components are randomly generated and will be different for different prompts: (1) the numbers in the demonstrations (e.g. `assert f(5, 8) == 13`) as long as they are correct for addition; (2) the name of the target function (e.g. `knk`, `ovs`).

```
assert knk([], 2) == [2]
assert knk([1], 2) == [1, 2]
assert knk([1, 3], 2) == [1, 3, 2]
assert knk([1, 3, 5], 2) == [1, 3, 5, 2]
assert knk([1, 3, 5, 7], 2) == [1, 3, 5, 7, 2]
assert knk([1, 3, 5, 7, 9], 2) == [1, 3, 5, 7, 9, 2]
assert knk([1, 3, 5, 7, 9, 11], 2) == [1, 3, 5, 7, 9, 11, 2]
def sum_list(l):
    return sum(l)
List 8 test cases of the above function sum_list, one in each line:
assert sum_list([]) == 0
assert sum_list([1]) == 1
assert sum_list([1, 3]) == 4
assert sum_list([1, 3, 5]) == 9
assert sum_list([1, 3, 5, 7]) == 16
assert sum_list([1, 3, 5, 7, 9]) == 25
assert sum_list([1, 3, 5, 7, 9, 11]) == 36
def sublist(l, sub):
    assert type
```

```
assert ovs([], 1) == [1]
assert ovs([2], 1) == [1, 2]
assert ovs([1, 2], 1) == [1, 1, 2]
assert ovs([1, 2], 3) == [1, 2, 3]
assert ovs([1, 2], 0) == [0, 1, 2]
assert ovs([1, 2, 3], 4) == [1, 2, 3, 4]
assert ovs([], 0) == [0]
assert ovs([1, 2], 0) == [0, 1, 2]
```

*Table 8.* Two example generations by CodeLlama corresponding to the prompts in Table 7. Note that both generations are flawed: (1) the model only generated 7 test cases instead of 8, even though the prompt requested 8. Then, it generated irrelevant contents, starting from `def sum_list(l):` (2) more than one generated test cases were wrong (e.g. in `assert ovs([2], 1) == [1, 2]`, the correct right-hand-side should be `[2, 1]`). More generally, we implemented a rule-based parser to analyze model generations and identify the error type (if any), and locate the first position of error.

B.2.2. BASELINES

We extensively tuned the hyperparameters in common baseline decoding algorithms, including

- nucleus sampling (Holtzman et al., 2020): we grid-searched top_p $\in [0.0, 0.7, 0.8, 0.9, 0.95, 1.0]$.

- argmax greedy decoding: equivalent to top_p = 0.0.

- standard autoregressive sampling: equivalent to top_p = 1.0.

- temperature scaling (Ackley et al., 1985): we grid-searched temperature $\in [0.0, 0.2, 0.4, 0.6, 0.8, 1.0, 1.2]$ (for each top_p).

Through the above grid search, we found that the best combination was top_p $= 0.95$, temperature $= 1.0$.

Besides, we consider baselines based on the *block-best-of-n* rejection sampling approach to incorporate process rewards. More details about this baseline are provided in the "Block verifier" part of Appendix B.2.3.

- block-best-of-n: we grid-searched $n \in [2, 4, 8]$, fixing the best combination of top_p and temperature found by the grid search above.

We will show that Tokenwise rejection sampling with backtracking (Algorithm 1) clearly outperforms all these baselines in terms of the quality vs. query complexity trade-off.

| layer index | $\text{Acc}_{\text{distinct}} \pm$ **std err** |
|---|---|
| 27 | $0.714 \pm 0.011$ |
| 28 | $0.711 \pm 0.016$ |
| 26 | $0.708 \pm 0.018$ |
| 30 | $0.706 \pm 0.036$ |
| 24 | $0.701 \pm 0.033$ |
| 31 | $0.688 \pm 0.028$ |
| 29 | $0.676 \pm 0.021$ |
| 25 | $0.672 \pm 0.030$ |
| 23 | $0.709 \pm 0.017$ |
| 3 | $0.700 \pm 0.028$ |
| 15 | $0.700 \pm 0.028$ |
| 19 | $0.692 \pm 0.028$ |
| 7 | $0.691 \pm 0.031$ |
| 11 | $0.650 \pm 0.041$ |
| ablation: random verifier | $0.663 \pm 0.027$ |
| baseline: nucleus sampling + temperature scaling | $0.660 \pm 0.042$ |

*Table 9.* Ablation: layer 27 representations of CodeLlama outperform layer 31 (the last layer) in terms of the quality of the error predictor trained based on these features. We control all other setting to be the same as the top-performing settings of the baseline (nucleus sampling top_p = 0.95 (Holtzman et al., 2020) and temperature 1.0), whose performance is also included in the table. The other rows in this table (layer 27 and layer 31) refer to applying Tokenwise rejection sampling with backtracking (Algorithm 1) using backtrack quota $Q = 4$, backtrack stride $B = 4$, and verifiers trained on layers 24, ..., 31 of the generator (CodeLlama), respectively. The row *ablation: random verifier* refers to a verifier that returns Uniform[0, 1], and uses the same $Q$, $B$ as the above. The experiment was repeated 5 times, and we report the standard errors. The rows are sorted by mean $\text{Acc}_{\text{distinct}}$ (Section 5.2.1).

### B.2.3. TRAINING THE VERIFIER

We follow almost the same training approach as Section 5.1.2. The differences are described below. The generator language model LM is a pretrained CodeLlama (Roziere et al., 2023) (7B parameters), which we freeze during our experiments.

**An intermediate layer provides more informative representations for verifier training than the last layer.** Instead of training the verifier V on top of the last layer (i.e. layer 31) representations of LM, we instead treat the layer index as a hyperparameter, and conducted a grid search over layer index $\in \{3, 7, 11, 15, 19, 23, 27, 31\}$. Among these candidates, layer 27 representations resulted in the best accuracy. We therefore exclusively used layer 27 representations in subsequent experiments, and finally conducted an ablation study on the top-performing setting of the baseline to back-test the impact of using other layers. Table 9 shows that layer 27 outperforms layer 31. We conjecture that the layer 31 representations may be too specific for the next-token prediction task, which is not necessarily the optimal for discriminating correct prefixes vs. incorrect ones. [12] We also include results for a few other layers near the final layer. Note that even with a sub-optimally chosen layer, the accuracy of Tokenwise rejection sampling with backtracking (Algorithm 1) still outperforms the top-performing settings of the baseline found through grid search. [13]

**With limited backtrack quota, it is better to be conservative in its usage.** The verifier V is trained with binary labels (1 if correct, 0 if wrong). Although there are a roughly equal number of training samples whose labels are 0 or are 1, using 0.5 as the error prediction threshold turned out to be suboptimal. Since our Tokenwise rejection sampling with backtracking (Algorithm 1) only allows a small backtrack quota $Q = 4$, it makes sense to only use backtrack quota when the error predictor is very confident that the current intermediate generation is wrong. Moreover, compared with our synthetic Dyck grammar setting (target length = 32) (Section 5.1), our code generation setting allows much longer generations (up to 384), which further justifies conservatively spending the small backtrack quota $Q$. Consequently, we consider decreasing the error prediction threshold to 0.1. Table 10 shows that 0.1 is a better error prediction threshold than the default 0.5 in all settings we tried.

---

[12]This is in line with some prior works that also observed that the final layers of language models tend to be more task-specific than the intermediate layers (Liu, 2019; Kovaleva et al., 2019; Rogers et al., 2021).

[13]See Appendix B.2.2 for details about baselines.

| $Q$ | $B$ | top_p | temperature | error prediction threshold | Acc$_{distinct}$ $\pm$ std err |
|---|---|---|---|---|---|
| 4 | 4 | 0.95 | 1.0 | 0.1 | $0.714 \pm 0.011$ |
| 4 | 4 | 0.95 | 1.0 | 0.5 | $0.676 \pm 0.019$ |
| 4 | 4 | 1.0 | 1.0 | 0.1 | $0.639 \pm 0.061$ |
| 4 | 4 | 1.0 | 1.0 | 0.5 | $0.604 \pm 0.047$ |
| 4 | 4 | 1.0 | 1.2 | 0.1 | $0.440 \pm 0.026$ |
| 4 | 4 | 1.0 | 1.2 | 0.5 | $0.334 \pm 0.013$ |
| 4 | 10 | 1.0 | 1.0 | 0.1 | $0.622 \pm 0.046$ |
| 4 | 10 | 1.0 | 1.0 | 0.1 | $0.604 \pm 0.030$ |

*Table 10.* Ablation: 0.1 is a better error prediction threshold than the default 0.5 in all settings we tried, including various nucleus sampling (Holtzman et al., 2020) top_p, temperature scaling, and backtrack stride $B$. In this table, we divide the rows into groups of 2, separated by double horizontal lines, such that within each group, the only difference is the error prediction threshold. In all groups, 0.1 leads to higher Acc$_{distinct}$ than 0.5. The experiment was repeated 5 times, and we report the standard errors.

| verifier # MLP layers | verifier validation accuracy | Acc$_{distinct}$ $\pm$ std err |
|---|---|---|
| 1 | 0.96 | $0.714 \pm 0.011$ |
| 4 | 0.97 | $0.699 \pm 0.038$ |
| 2 | 0.97 | $0.687 \pm 0.035$ |
| 8 | 0.97 | $0.684 \pm 0.015$ |
| ablation: random verifier | 0.50 | $0.663 \pm 0.027$ |
| baseline: nucleus sampling + temperature scaling | N/A | $0.660 \pm 0.042$ |

*Table 11.* Ablation: Deeper verifiers do not outperform the 1-linear-layer verifier even though they can be trained to similar *error-predicting* accuracies on held-old validation set. We control all other setting to be the same as the top-performing settings of the baseline (nucleus sampling top_p = 0.95 (Holtzman et al., 2020) and temperature 1.0), whose performance is also included in the table. The other rows in this table refer to applying Tokenwise rejection sampling with backtracking (Algorithm 1) using backtrack quota $Q = 4$, backtrack stride $B = 4$, and verifiers with 1, 2, 4, 8 layers, respectively. The row *ablation: random verifier* refers to a verifier that returns Uniform[0, 1], and uses the same $Q$, $B$ as the above. The experiment was repeated 5 times, and we report the standard errors. The rows are sorted by mean Acc$_{distinct}$ (Section 5.2.1).

**Block verifier.** Our verifier applies to the token level, i.e. predicting an accept/reject action after the generator LM generates each token. In many practical settings (including ours), it is natural to divide the generated output into *blocks* (each block may contain multiple tokens), e.g. in writing math proofs, each block may correspond to one reasoning step; in writing codes, each block may correspond to one line of codes. Recent works achieved strong empirical performance by generating multiple candidates for each block of intermediate model generations, train process reward models that evaluate each candidate, and select the best-scoring candidate (see e.g. Wu et al. (2024) and references therein). We refer to this as the "block-best-of-n" approach. To compare with such "block-best-of-n" baselines, we train "block verifiers" V$_{block}$ which scores prefixes that are full lines of model output for our task. We will show that this "block best-of-n" approach is helpful, but is outperformed by our Tokenwise rejection sampling with backtracking (Algorithm 1) in terms of accuracy-efficiency trade-off.

**Does a deeper verifier perform better?** The above experiments follow Section 5.1.2 in training a single-linear-layer verifier. In this section, we test the effects of scaling up the verifier depth. Specifically, we test verifiers based on Multi-Layer Perceptrons (Rosenblatt, 1958) of depths 2, 4, 8, with ReLU activations (Nair & Hinton, 2010) between adjacent parameterized layers. Table 11 shows that more MLP layers did not outperform the 1-linear-layer verifier even though they can be trained to similar *error-predicting* accuracies, measured by their accuracy in predicting whether a prefix is correct or incorrect on a held-old validation set of prompts for our task (Section 5.2.1) followed by partial generations by CodeLlama. In other sections of this paper, unless otherwise noted, we always use a single-linear-layer verifier for Tokenwise rejection sampling with backtracking (Algorithm 1) (and of course, no verifier for baselines (Appendix B.2.2) ).

**Where are the potentials for further improving Acc$_{distinct}$?** How optimal are our verifiers, and what are some ways to further improve them? To probe these potentials, we wrote a rule-based groundtruth verifier for our task (Section 5.2.1) and used it as a drop-in replacement of our trained verifier. Table 12 shows that the Acc$_{distinct}$ enabled by our trained verifier

almost reached the $\text{Acc}_{\text{distinct}}$ enabled by the groundtruth verifier, showing that improving verifier training may not be the most fruitful direction for further improvement. Interestingly, using a much larger $Q$ or $B$ (increasing from 4 to 10) does not necessarily improve the accuracy (sometimes even *decreasing* the accuracy). We conjecture that in these experiments, the (imperfect) generator oracle (CodeLlama), not the verifier, was the bottleneck for $\text{Acc}_{\text{distinct}}$. As a result, unnecessarily backtracking and forcing the model to re-generate more tokens may increase the chance that the model makes mistakes.

| verifier type | $Q$ | $B$ | $\text{Acc}_{\text{distinct}} \pm$ **std err** |
|---|---|---|---|
| groundtruth | 4 | 4 | $0.719 \pm 0.022$ |
| groundtruth | 10 | 4 | $0.717 \pm 0.015$ |
| trained | 4 | 4 | $0.714 \pm 0.011$ |
| trained | 10 | 4 | $0.692 \pm 0.025$ |
| ablation: random verifier | 4 | 4 | $0.663 \pm 0.027$ |
| baseline: nucleus sampling + temperature scaling | 0 | 0 | $0.660 \pm 0.042$ |
| trained | 4 | 10 | $0.622 \pm 0.046$ |

*Table 12.* Ablation: Our trained verifier approaches the accuracy of the groundtruth verifier, evaluated by their ability to assist CodeLlama in completing our test case generation task (Section 5.2.1) using Tokenwise rejection sampling with backtracking (Algorithm 1). In these experiments, we control the nucleus sampling (Holtzman et al., 2020) top_p = 0.95 and temperature scaling = 1.0 which are the optimal setting for baseline, found by grid search (Appendix B.2.2). The rows are sorted by $\text{Acc}_{\text{distinct}}$. The row *ablation: random verifier* refers to a verifier that returns Uniform[0, 1]. Interestingly, using a much larger $Q$ or $B$ does not necessarily improve the accuracy (sometimes even *decreasing* the accuracy). We conjecture that the generator model, CodeLlama, is imperfect, so unnecessarily backtracking and forcing the model to re-generate more tokens may increase the chance that the model makes mistakes. The experiment was repeated 5 times, and we report the standard errors.

B.2.4. FULL RESULTS OF CODELLAMA EXPERIMENTS IN SECTION 5.2

**Caption for Table 13 (on the next page).**   Tokenwise rejection sampling with backtracking (Algorithm 1) improves accuracy and outperforms commonly used baselines, including nucleus sampling top_p, temperature scaling (temp), and block best-of-n (BBoN) (Appendix B.2.3). Baselines are extensively hyperparameter tuned (Appendix B.2.2). Backtrack quota $Q = 0$ means a baseline without verifier. When $Q > 0$, the row denotes Algorithm 1 with the corresponding $Q$ and $B$. The column *layer idx* denotes which layer of CodeLlama provided the representations for training the verifier, and *err threshold* denotes the cutoff below which the verifier output is interpreted as a rejection (both were experimented in Appendix B.2.3). When BBoN is specified, the row denotes the number of candidates generated for each block; otherwise, the row does not use block best-of-n. The rows are sorted by $\text{Acc}_{\text{distinct}}$. Controlling top_p and temperature, Algorithm 1 leads to better tradeoff between $\text{Acc}_{\text{distinct}}$ and query complexity $\mathcal{C}$ (both defined in Section 5.2.1) than all other methods. The experiment was repeated 5 times, and we report the standard errors.

| $Q$ | $B$ | layer idx | err threshold | top_p | temp | BBoN | $\text{Acc}_{\text{distinct}} \pm$ std err | $\mathcal{C}$ |
|---|---|---|---|---|---|---|---|---|
| 4 | 4 | 27 | 0.1 | 0.95 | 1.0 | | $0.714 \pm 0.011$ | $39443 \pm 235$ |
| 4 | 4 | 31 | 0.5 | 0.95 | 1.0 | | $0.688 \pm 0.028$ | $39629 \pm 135$ |
| 0 | | 27 | | 0.95 | 1.0 | 2 | $0.684 \pm 0.038$ | $39364 \pm 1252$ |
| 4 | 4 | 31 | 0.1 | 0.95 | 1.0 | | $0.677 \pm 0.033$ | $39546 \pm 98$ |
| 4 | 4 | 27 | 0.5 | 0.95 | 1.0 | | $0.676 \pm 0.019$ | $38555 \pm 140$ |
| 0 | | | | 0.95 | 1.0 | | $0.660 \pm 0.042$ | $38231 \pm 165$ |
| 4 | 4 | 27 | 0.1 | 1.0 | 1.0 | | $0.639 \pm 0.061$ | $31274 \pm 1559$ |
| 0 | | | | 0.9 | 1.0 | | $0.634 \pm 0.023$ | $38393 \pm 14$ |
| 0 | | | | 0.9 | 1.2 | | $0.630 \pm 0.028$ | $38005 \pm 232$ |
| 0 | | | | 0.8 | 1.2 | | $0.627 \pm 0.015$ | $38343 \pm 90$ |
| 0 | | 27 | | 0.95 | 1.0 | 4 | $0.623 \pm 0.036$ | $65496 \pm 7638$ |
| 4 | 10 | 27 | 0.1 | 1.0 | 1.0 | | $0.622 \pm 0.046$ | $32923 \pm 1772$ |
| 4 | 4 | 27 | 0.5 | 1.0 | 1.0 | | $0.604 \pm 0.047$ | $31091 \pm 968$ |
| 4 | 10 | 27 | 0.5 | 1.0 | 1.0 | | $0.604 \pm 0.030$ | $27287 \pm 7580$ |
| 0 | | | | 0.95 | 1.2 | | $0.584 \pm 0.027$ | $36601 \pm 535$ |
| 0 | | | | 1.0 | 0.8 | | $0.562 \pm 0.021$ | $36610 \pm 669$ |
| 0 | | 27 | | 0.95 | 1.0 | 8 | $0.559 \pm 0.038$ | $122933 \pm 3832$ |
| 0 | | | | 0.7 | 1.2 | | $0.531 \pm 0.035$ | $38400 \pm 0$ |
| 0 | | | | 0.95 | 0.8 | | $0.523 \pm 0.029$ | $38386 \pm 28$ |
| 0 | | | | 0.8 | 1.0 | | $0.511 \pm 0.028$ | $38400 \pm 0$ |
| 0 | | | | 1.0 | 1.0 | | $0.504 \pm 0.025$ | $30754 \pm 1272$ |
| 0 | | | | 0.9 | 0.8 | | $0.466 \pm 0.032$ | $38400 \pm 0$ |
| 4 | 4 | 27 | 0.1 | 1.0 | 1.2 | | $0.440 \pm 0.026$ | $24916 \pm 954$ |
| 0 | | | | 1.0 | 0.6 | | $0.399 \pm 0.070$ | $38320 \pm 73$ |
| 0 | | | | 0.7 | 1.0 | | $0.353 \pm 0.021$ | $38400 \pm 0$ |
| 0 | | | | 0.8 | 0.8 | | $0.351 \pm 0.039$ | $38400 \pm 0$ |
| 0 | | | | 0.95 | 0.6 | | $0.337 \pm 0.053$ | $38400 \pm 0$ |
| 4 | 4 | 27 | 0.5 | 1.0 | 1.2 | | $0.334 \pm 0.013$ | $24217 \pm 1214$ |
| 0 | | | | 0.9 | 0.6 | | $0.284 \pm 0.044$ | $38400 \pm 0$ |
| 0 | | | | 1.0 | 1.2 | | $0.269 \pm 0.025$ | $21906 \pm 1780$ |
| 0 | | | | 0.7 | 0.8 | | $0.239 \pm 0.019$ | $38400 \pm 0$ |
| 0 | | | | 0.8 | 0.6 | | $0.212 \pm 0.011$ | $38400 \pm 0$ |
| 0 | | | | 1.0 | 0.4 | | $0.207 \pm 0.029$ | $38400 \pm 0$ |
| 0 | | | | 0.95 | 0.4 | | $0.176 \pm 0.013$ | $38400 \pm 0$ |
| 0 | | | | 0.9 | 0.4 | | $0.147 \pm 0.013$ | $38400 \pm 0$ |
| 0 | | | | 0.7 | 0.6 | | $0.101 \pm 0.028$ | $38400 \pm 0$ |
| 0 | | | | 1.0 | 0.2 | | $0.080 \pm 0.020$ | $38400 \pm 0$ |
| 0 | | | | 0.8 | 0.4 | | $0.074 \pm 0.027$ | $38400 \pm 0$ |
| 0 | | | | 0.95 | 0.2 | | $0.057 \pm 0.018$ | $38400 \pm 0$ |
| 0 | | | | 0.9 | 0.2 | | $0.029 \pm 0.015$ | $38400 \pm 0$ |
| 0 | | | | 0.7 | 0.4 | | $0.025 \pm 0.016$ | $38400 \pm 0$ |
| 0 | | | | 0.8 | 0.2 | | $0.021 \pm 0.014$ | $38400 \pm 0$ |
| 0 | | | | 0.7 | 0.2 | | $0.018 \pm 0.011$ | $38400 \pm 0$ |
| 0 | | | | 0.0 | 1.0 | | $0.013 \pm 0.000$ | $38400 \pm 0$ |

*Table 13.* Tokenwise rejection sampling with backtracking (Algorithm 1) improves accuracy and outperforms commonly used baselines, including nucleus sampling top_p, temperature scaling (temp), and block best-of-n (BBoN) (Appendix B.2.3). Due to space constraints, more detailed captions are in the beginning of this section. ***To help readers parse these results, we included smaller tables, each analyzing a single aspect:*** please refer to Table 4 in Section 5.2.2, Table 10 in Appendix B.2.3, Table 9 in Appendix B.2.3, and Figure 3 in Section 5.2.3.

B.2.5. VISUALIZING THE QUERY EFFICIENCY OF TOKENWISE REJECTION SAMPLING WITH BACKTRACKING

This section complements the query efficiency visualization discussed in Section 5.2.3. [14]

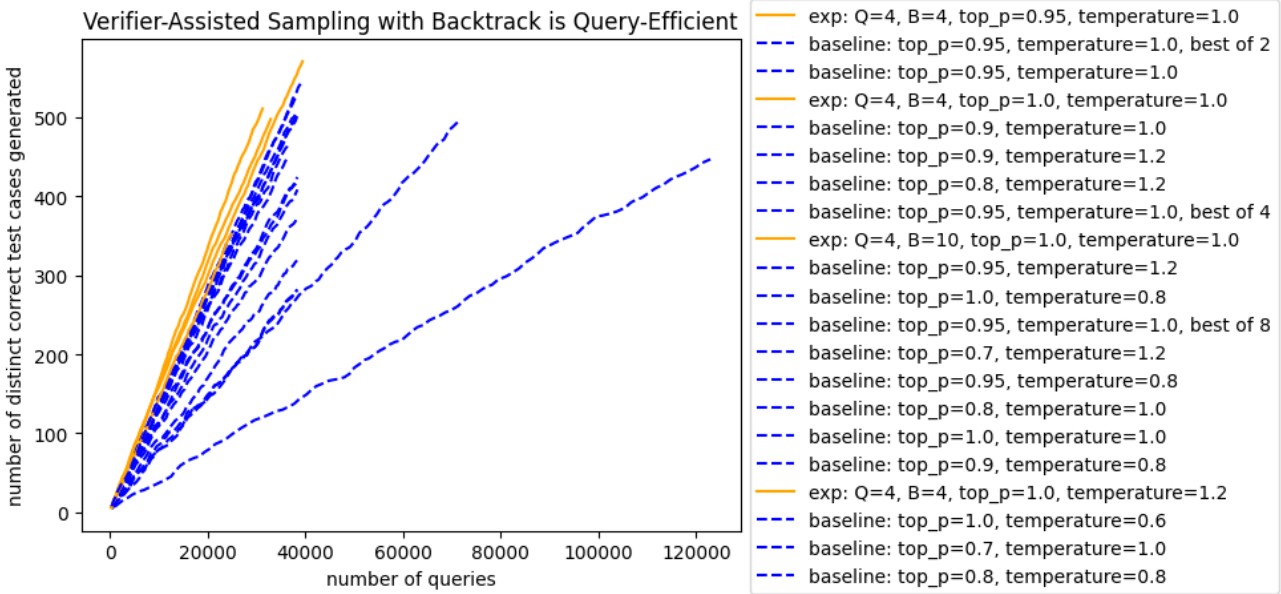

*Figure 3.* Tokenwise rejection sampling with backtracking (Algorithm 1) is query-efficient. The horizontal axis denotes query complexity $\mathcal{C}$, and the vertical axis denotes the number of distinct correct test cases generated $N_{\text{distinct correct}}$, both defined in Section 5.2.1. Blue dashed lines correspond to the baselines (described in Appendix B.2.2), whereas orange solid lines correspond to Tokenwise rejection sampling with backtracking with various $Q$ and $B$, both defined in Algorithm 1. Since the slopes of the orange curves are visibly greater than the slopes of the blue curves, we conclude that Tokenwise rejection sampling with backtracking is more query-efficient than baselines. The experiment was repeated 5 times, and each dot is the average metric of these 5 runs. The specific numbers and standard errors are reported in Table 13. A more zoomed-in version of this plot is in Figure 4.

---

[14]This visualization here in Figure 3 slightly favors the "block best-of-n sampling" baseline, because its implementation stops the decoding process once the requested number of test cases are generated, whereas when running our algorithm or non-best-of-n baselines, the model is allowed to (and in fact does indeed) generate irrelevant tokens afterwards, which hurts query complexity. Even under this disadvantage, Tokenwise rejection sampling with backtracking still outperforms the "block best-of-n sampling" baselines.

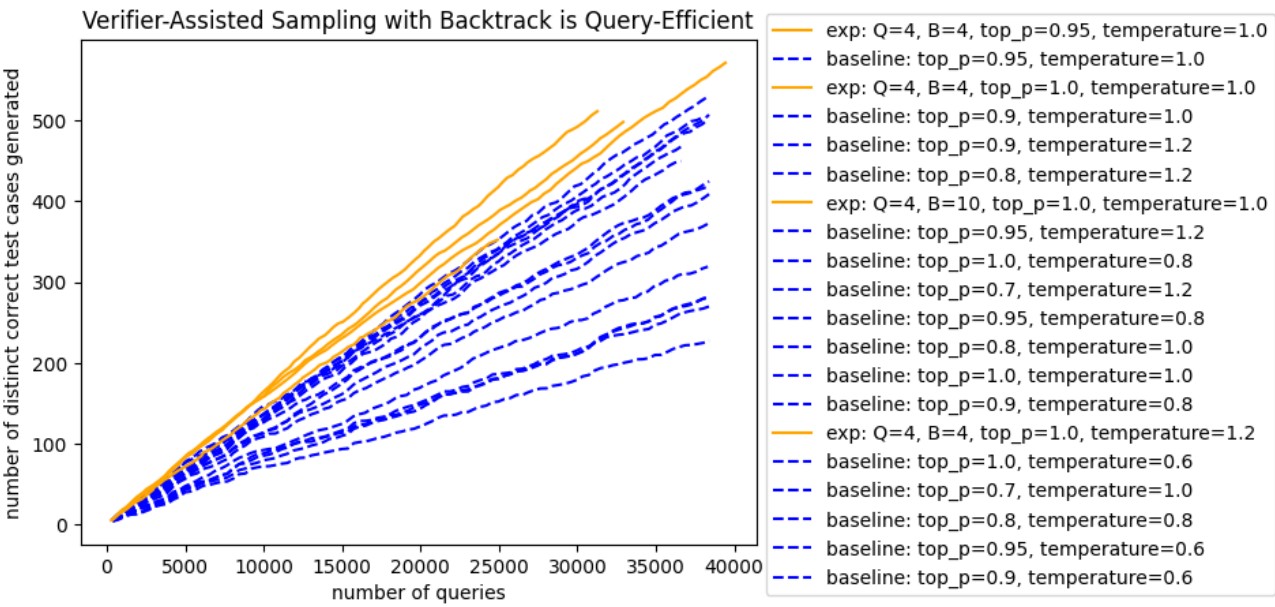

*Figure 4.* Similar to Figure 3, just more zoomed-in, excluding block best-of-n baselines (Appendix B.2.3).

B.2.6. TOKENWISE REJECTION SAMPLING WITH BACKTRACKING GENERALIZES BETTER TO OUT-OF-DISTRIBUTION
     PROMPTS

In this section, we show that Tokenwise rejection sampling with backtracking (Algorithm 1) generalizes better to out-of-distribution prompts than the best nucleus sampling and temperature scaling baseline in Appendix B.2.2. Unlike the synthetic Dyck grammar setting, on real-world LLMs we do not have a precise quantitative control over how "out-of-distribution" a prompt is for the LLM. We therefore assume that a sufficient condition for a prompt in our setup to be out-of-distribution is that the name of the target function denotes some meaning which is different from the actual implemented functionality (i.e. list append) (recall the task setup in Section 5.2.1). Two examples of such out-of-distribution prompt are provided in Table 14. We validate this assumption by observing that the accuracy indeed degrades on such "out-of-distribution" prompts, suggesting that the model is indeed confused by the inconsistency between the function names and the function implementations. However, analogous to our observations on the synthetic Dyck grammar (Section 5.1.5), Tokenwise rejection sampling with backtracking (Algorithm 1) again suffers much less reduction in accuracy on these "out-of-distribution" prompts. The detailed comparisons are reported in Table 15.

---

```
def f(a, b):
      return a + b

List 8 test cases of the above function f, one in each line:
assert f(6, 5) == 11
assert f(3, 2) == 5
assert f(5, 4) == 9
assert f(1, 5) == 6
assert f(5, 4) == 9
assert f(3, 5) == 8
assert f(5, 6) == 11
assert f(2, 6) == 8

def add(l, item):
      assert type(l) is list
      l.append(item)
      return l

List 8 test cases of the above function add, one in each line:
```

---

```
def f(a, b):
      return a + b

List 8 test cases of the above function f, one in each line:
assert f(8, 7) == 15
assert f(8, 1) == 9
assert f(4, 7) == 11
assert f(8, 4) == 12
assert f(7, 4) == 11
assert f(8, 4) == 12
assert f(1, 1) == 2
assert f(5, 5) == 10

def exp(l, item):
      assert type(l) is list
      l.append(item)
      return l

List 8 test cases of the above function exp, one in each line:
```

---

Table 14. Two example *out-of-distribution* prompts for generating test cases for a simple implementation of the append function for Python lists. Different from the prompts in Table 7, here the function names denote a clear meaning (e.g. add or exp), which, however, is different from what the function implements (i.e. append).

| $Q$ | $B$ | err threshold | in-distribution Acc$_{\text{distinct}} \pm$ std err | OOD Acc$_{\text{distinct}} \pm$ std err |
|---|---|---|---|---|
| 4 | 4 | 0.1 | $0.714 \pm 0.011$ | $0.710 \pm 0.029$ |
| 4 | 4 | 0.5 | $0.676 \pm 0.019$ | $0.687 \pm 0.024$ |
| 0 | | | $0.660 \pm 0.042$ | $0.606 \pm 0.034$ |

*Table 15.* Tokenwise rejection sampling with backtracking (Algorithm 1) generalizes better to out-of-distribution prompts than the best nucleus sampling and temperature scaling baseline in Appendix B.2.2, which we identified by grid search (Table 13) to be top_p = 0.95, and temperature = 1.0. We manually pick 10 target function names according to Appendix B.2.6 which were unseen when training the verifier (Appendix B.2.3). When backtrack quota $Q = 0$, the row denotes a baseline algorithm that does not use the verifier (and consequently the backtrack stride $B$ will not matter). The column *err threshold* denotes the cutoff below which the error predictor output is interpreted as a rejection (Appendix B.2.3). When $Q > 0$, the row denotes Tokenwise rejection sampling with backtracking (Algorithm 1) with the corresponding $Q$ and $B$. Tokenwise rejection sampling with backtracking (Algorithm 1) suffered minor or no drop between in-distribution and OOD Acc$_{\text{distinct}}$, whereas the baseline suffered a drop by more than one standard error. The experiment was repeated 5 times, and we report the standard errors.

## B.3. Additional ablation experiments on the Tokenwise rejection sampling with backtracking algorithm (Algorithm 1)

Besides the ablation experiments in Appendix B.2.3 which probe various aspects of verifier training, in this section, we focus on one algorithmic component.

Concretely, line 10 of Tokenwise rejection sampling with backtracking (Algorithm 1) re-generates the erased positions using argmax. This was motivated by our results in Section 5.1.1 which suggest that *out-of-distribution prefix* is a cause of generator mistakes. As a remedy, redoing the erased positions using argmax is intended to increase the generator-predicted probability of the partially sampled generation, which (concatenated with the prompt) will be the prefix for subsequent generation steps. We include an ablation study verifying that this improves the accuracy, significantly under the synthetic data setting (Table 16), and only slightly (without hurting diversity) under the real data setting (Table 17).

| sampling algorithm | #errors $\pm$ std err |
|---|---|
| Algorithm 1 | $179.4 \pm 1.020$ |
| ablation: no argmax | $245.8 \pm 8.658$ |

*Table 16.* Re-generating the erased positions using argmax in Tokenwise rejection sampling with backtracking (Algorithm 1) reduces completion errors on unseen out-of-distribution (OOD) prefixes in Dyck grammar. We fixed nucleus sampling (Holtzman et al., 2020) top_p = 0.9, backtrack quota $Q = 4$, and backtrack stride $B = 4$ (the best settings in Table 3). The row "ablation: no argmax" refers to removing lines 9-12 in Algorithm 1. We report the number of completion errors that occur when completing an unseen set of 10000 independently sampled out-of-distribution prompts $\text{Dyck}_{OOD}^{unseen}$. The experiment was repeated 5 times, and we report the standard errors.

| sampling algorithm | err threshold | $\text{Acc}_{\text{distinct}} \pm$ std err |
|---|---|---|
| Algorithm 1 | 0.1 | $0.714 \pm 0.011$ |
| ablation: no argmax | 0.1 | $0.711 \pm 0.032$ |
| Algorithm 1 | 0.5 | $0.676 \pm 0.019$ |
| ablation: no argmax | 0.5 | $0.663 \pm 0.023$ |

*Table 17.* Re-generating the erased positions using argmax in Tokenwise rejection sampling with backtracking (Algorithm 1) slightly improves the accuracy-diversity tradeoff (Section 5.2.1) in our test case generation task. We fixed nucleus sampling (Holtzman et al., 2020) top_p = 0.95, backtrack quota $Q = 4$, and backtrack stride $B = 4$ (the best settings in Table 13). The row "ablation: no argmax" refers to removing lines 9-12 in Algorithm 1. The column *err threshold* denotes the cutoff below which the error predictor output is interpreted as a rejection (Appendix B.2.3). The experiment was repeated 5 times, and we report the standard errors.

