# OpenReview forum: "On the Query Complexity of Verifier-Assisted Language Generation"
_ICML.cc/2025/Conference — ICML 2025 poster_

### Official Review · Reviewer_LdiQ · 2025-03-14

**Overall Recommendation:** 3

**Summary:**

This paper introduces a mathematical framework for analyzing verifier-assisted language generation and shows that process verifiers can transform intractable generation problems into tractable ones.  Through theoretical analysis and experiments on synthetic grammar and code generation tasks, they demonstrate their approach significantly outperforms baselines in computational efficiency, accuracy, and generation diversity.

**Claims And Evidence:**

Yes

**Essential References Not Discussed:**

The paper covers most essential related work.

**Experimental Designs Or Analyses:**

1. The authors pretrain language models on synthetic data with controlled parameters, then test on out-of-distribution samples. They train lightweight verifiers on error cases and evaluate backtracking effectiveness. The design is sound, with proper controls and ablations on backtrack quota and stride length.
2. The authors evaluate their method against multiple baselines across metrics of accuracy, diversity, and query complexity. They conduct comprehensive hyperparameter searches and ablation studies.

**Methods And Evaluation Criteria:**

The methods and evaluation criteria in the paper are well-aligned with the problem of verifier-assisted language generation.

**Other Comments Or Suggestions:**

N/A

**Other Strengths And Weaknesses:**

Other Strengths:
1. The paper bridges theoretical analysis and practical implementation, showing both mathematical proofs of complexity benefits and empirical evidence of performance gains.
2. tokenwise rejection sampling with backtracking algorithm is relatively simple to implement, making it accessible for practical use while still providing significant benefits.
3. The error analysis in Appendix B.1.3 offers valuable insights into why certain errors persist, which could guide future improvements.

Other Weaknesses:

1. The real-world experiments focus only on code generation (specifically test case generation), which may not fully demonstrate the method's applicability to other constrained generation tasks like mathematical reasoning or formal specification.

**Questions For Authors:**

1. Have you explored how the approach performs when constraints are "softer" or probabilistic rather than binary?
2. For the CodeLlama experiment, you focus on test case generation for simple list append functions. How do you expect the results to generalize to more complex programming tasks with deeper logical dependencies and constraints?

**Relation To Broader Scientific Literature:**

1. Verifier-assisted generation relates to inference-time optimization methods like best-of-N sampling, and provides formal analysis of when and why verifiers help, moving beyond empirical scaling laws.
2. The tokenwise rejection sampling with backtracking approach extends classical rejection sampling techniques with a process verifier, connecting to both statistical sampling methods and search algorithms like beam search

**Theoretical Claims:**

1. Theorem 1 shows that without a verifier, certain constrained generation tasks require at least $2^{D-1}$ oracle queries in expectation. The proof constructs a scenario where the algorithm must identify a specific prefix among all possible prefixes and uses Yao's minimax lemma to establish the lower bound for randomized algorithms.
2. Theorem 2 establishes that even with simple constraints like those in the knapsack problem, constrained generation can be NP-hard without a verifier. The proof is a straightforward reduction from the knapsack problem.
3. Proposition 1 demonstrates the computational benefits of tokenwise rejection sampling with a process verifier compared to standard rejection sampling. This proof calculates expected queries for both methods on a simple example.
4. Proposition 2 shows that tokenwise rejection sampling doesn't sample from the restricted distribution. The proof provides a counterexample.

---

> ### Author Rebuttal · Authors · 2025-04-01
>
> Thank you for your review! In the following, we respond to a few points raised in your review.
>
> **1. When constraints are "softer" or probabilistic rather than binary.**
>
> There could be multiple types of constrained generation tasks with non-binary constraints. In the following, we explain our thoughts on two common types:
>
> (1) Suppose the constraints are specified as a probabilistic distribution over responses, such that better responses are assigned higher probabilities, and the task is to sample according to this distribution. The main difficulty then is understanding what are reasonable assumptions on the power of the process verifier. Namely, Proposition 2 suggests that a verifier that solely outputs a binary answer (whether there exists a completion of the prefix in the target domain) is not powerful enough. On the other hand, a simple modification of part 2 of Proposition 1 also shows that a “perfectly calibrated” process verifier (which outputs the conditional probability of the prefix according to the target distribution) can be trivially used to design a linear-time algorithm to sample from the target distribution. Finding reasonable assumptions would involve both theoretical reasoning and incorporating empirical evidence from trained verifiers.
>
> (2) Suppose the constraints are specified as scores over the responses, such that better responses are assigned higher scores, and the task is to sample responses with high scores. Under this setting, one approach to the verifier is known as reward modeling in the literature. (Our response to your second question below includes more discussion on reward modeling.) We surmise that our verifier-assisted approach can be applied to this setting as well, by either setting a cutoff for our verifier (reward model) predicted scores, or accepting a prefix according to a probability predicted by the verifier. These approaches can very likely improve the average reward of the final generated responses. Like case (1) above, one challenge is to maintain the calibration of the output distribution against some target distribution.
>
> **2. How do you expect the results to generalize to more complex programming tasks with deeper logical dependencies and constraints?**
>
> Thank you for raising this important question! A key component of our approach is the verifier – the success of our token-wise rejection sampling with backtracking algorithm relies on the ability of the verifier to correctly reject partially decoded sequences that are wrong. Therefore, more complex programming tasks may require larger verifiers or more verifier training data.
>
> Despite these challenges, prior works have shown strong potentials for verifiers. For example, one approach to training verifiers is to “soften” them as reward models: train the verifier to predict a higher score for correct responses, and lower scores for incorrect ones. In the case of programming tasks, the RewardBench leaderboard [1] shows that researchers have developed accurate reward models that can distinguish between correct code vs. buggy codes in several common programming languages (with >95% accuracy).
>
> It may be a more challenging task to train process verifiers (Definition 3 in our paper) that can check any partially generated codes at arbitrary positions. However, we believe that since code is highly structured, it is promising to incorporate existing tools such as compilers/interpreters to provide additional signals for verification, or to automatically break down the programming tasks in to logical steps (e.g. a simple baseline is to treat each full line as a step), and train the verifier to check at the end of each logical step.
>
> [1] Nathan Lambert et al, RewardBench: Evaluating Reward Models for Language Modeling, 2024.

---

### Official Review · Reviewer_jJPP · 2025-03-15

**Overall Recommendation:** 2

**Summary:**

The paper presents several theoretical and experimental results on the query complexity of constrained decoding. First, in Section 3, it demonstrates theoretically that there exists tasks for which constrained decoding is difficult without a verifier. Then, in Section 4, it shows that for certain problems, the access to a process verifier significantly makes constrained decoding easier. Finally, in Section 5, the paper presents experimental results to validate their theoretical results in Sections 3 and 4.

**Claims And Evidence:**

I feel that the theoretical results are very weak as they only show the existence of hard problems but do not touch the difficulty in general. Can the authors comment on this? Moreover, the results only consider cases where the members of the constraint set A have fixed length. In practical use cases, the target length can vary, e.g., for code syntax and JSON.

There is a discrepancy between the theoretical results and the experimental results. While the theoretical results assume a perfect verifier that outputs a boolean value, the experiments leverage imperfect, trained verifiers that output a probability. On a high level, the problems studied by the theoretical and the experimental sections can be interpreted differently: the former focuses on the generation of structured outputs that can be described by symbolic constraints, while the latter targets the generation of any outputs with a trained verifier. The accuracy of the imperfect verifiers is also low, e.g., ~70% as shown in Section B.2.3. I think it is possible to construct perfect verifiers for both experiments in Section 5, e.g., leveraging SynCode (https://arxiv.org/abs/2403.01632). Why train the verifiers instead?

**Essential References Not Discussed:**

N/A.

**Experimental Designs Or Analyses:**

The experiment results are not impressive either. If we compare the accuracy between Tables 1 and 2, the access to a verifier with Q=1 or Q=2 can even decrease the accuracy, for the same B value. I am not even sure if the results for Q=1 or Q=2 are interesting. Since we have access to the verifier, why do we only use it once or twice?

**Methods And Evaluation Criteria:**

The backtracking condition in Algorithm 1, i.e., Line 6, does not look correct to me. Shouldn’t it be V(x+s) = 0?

**Other Comments Or Suggestions:**

N/A.

**Other Strengths And Weaknesses:**

N/A.

**Questions For Authors:**

N/A.

**Relation To Broader Scientific Literature:**

I think the topic studied by this paper is very important, given LLMs’ tendency to hallucinate and the promise of using constrained decoding to address these hallucinations. However, I have various concerns about the paper’s results, as discussed in the other review sections.

**Theoretical Claims:**

The correctness of the theoretical claims look good to me. As discussed in the “Claims And Evidence” section, they are just not strong enough in my opinion.

---

> ### Author Rebuttal · Authors · 2025-04-01
>
> Thank you for your review! In the following, we respond to a few points raised in your review.
>
> **1. Scope of our theory, imperfect verifiers**
>
> We see one of our main theoretical contributions as being a new formalism for theoretically reasoning about algorithms for verifier-assisted language generation. Moreover, our theoretical analyses already illustrate a few practically relevant claims: the importance and power of verifiers from information-theoretical (Theorem 1) and computational (Theorem 2) perspectives, and the challenge of maintaining calibration while incorporating a verifier (Proposition 2).
>
> We agree with the reviewer that there are multiple ways in which our results should be extended in order to guide practical algorithm design—in particular, an important aspect is dealing with imperfect verifiers. The main challenge is mathematical modeling: namely, what are assumptions on the verifier that comport with empirical evidence (i.e. satisfied by trained verifiers in practice), but are also provably theoretically helpful, and conducive to designing better algorithms.
>
> We hope that our proposed theoretical framework and these initial results will motivate the community to tackle such questions. Finally, for some of these relaxations of the theoretical assumptions, our paper includes careful empirical analyses which will hopefully inspire future theory.
>
> **2. Adding experimental results with perfect verifiers**
>
> Thank you for your suggestion! We ran experiments using perfect groundt-truth verifiers. We show the results for the Python test case generation task in this figure: https://ibb.co/pv5z26Wk. In particular, our trained verifier approaches the accuracy of the ground-truth verifier.
>
> In addition to perfect verifiers, we also reported experimental results with imperfect, trained verifiers with an eye towards other realistic tasks (e.g. creative story writing), because in such settings it’s not straightforward to implement perfect verifiers. This is to illustrate that our approach of training neural network-based verifiers is applicable to more general tasks.
>
> Our Dyck grammar task can be seen as a special case of the task proposed in SynCode. We will cite this paper and discuss its relevance in our Related Works section.
>
> **3. Why not use larger Q, B; can larger Q, B decrease the accuracy?**
>
> We first note a possible misunderstanding: Table 1 and Table 2 are not directly comparable because they test different settings. In Table 1, we use a ground-truth verifier to exactly locate the first position of error, and allow the generator to backtrack once. In Table 2, we use a trained verifier.
>
> In particular, Table 2 shows that, when backtrack quota (Q) and stride (B) are small, increasing them always results in higher accuracy.
>
> When using much larger Q or B, our results in https://ibb.co/pv5z26Wk show that even with a perfect verifier, applying it too many times can decrease the accuracy. We conjecture that the generator model, CodeLlama, is imperfect, so unnecessarily backtracking and forcing the model to re-generate more tokens may increase the chance that the model makes mistakes.
>
> **4. Algorithm 1, i.e., Line 6**
>
> Yes it should be V(x+s) = 0. Thank you for catching this typo!

---

### Official Review · Reviewer_BPwH · 2025-03-17

**Overall Recommendation:** 5

**Summary:**

This work studies the problem of constrained autoregressive language generation from the perspective of computational complexity, showing that for even very simple autoregressive oracles (i.e. autoregressive language models) and very simple constraints, the task of constrained generation can be computationally hard, while becoming tractable if given access to a process verifier. Instead of trying to propose some new constrained decoding algorithms, this paper aims to conduct an a systematic study (theoretical and empirical) covering three most commonly leveraged constrained decoding paradigms: (1) sequence-level (rejection) sampling w/ access to a sequence-level verifier (2) token-level rejection sampling with a process verifier (that works on partial sequences) (3) token-level rejection sampling + backtracking. The paper first (i) establishes the difficulty of constrained generation due to the oracle itself being complicated, then (ii) demonstrates the exponential separation between (1) and (2) when the oracle is simple, and lastly (iii) verifies the advantages of (3) over (2), i.e. the benefit of backtracking, as well as the advantages of (2) over (1) with well-controlled synthetic experiments. The theoretical results in this paper are not ground-breaking but the arguments and ideas behind are somewhat fundamental to the study of constrained autoregressive generation in general. In addition, Proposition 2 also makes a nice and clean argument uncovering the bias (in terms of approximate probabilistic inference) carried by some of the commonly adopted constrained generation approaches for structured output (i.e. JSON generation).

**Claims And Evidence:**

The claims are well supported by either theoretical results or empirical evidences.

**Essential References Not Discussed:**

The authors should probably include references to some of the controllable text generation literature such as *Yang, Kevin, and Dan Klein. "FUDGE: Controlled text generation with future discriminators." arXiv preprint arXiv:2104.05218 (2021).* and more.

**Experimental Designs Or Analyses:**

Yes.

**Methods And Evaluation Criteria:**

The synthetic empirical evaluation employed by this paper is very rigorous and immediately to-the-point.

**Other Comments Or Suggestions:**

N/A

**Other Strengths And Weaknesses:**

See summary.

**Questions For Authors:**

1. It would be nice if the authors/other people can study this problem in the settings where proper conditioning, i.e. we want to sample from $p(s) \propto \mathbf{1}(s \in A) p_{\mathcal{O}}(s)$, is desired.
2. For the models trained on Dyck grammar, as you sample from them, how many samples to do you need to cover the whole support of the constraints? More specifically, do they achieve near 0 KL divergence between the ground-truth and the trained model?

**Relation To Broader Scientific Literature:**

See summary.

**Theoretical Claims:**

Yes, I checked the (sketch) proofs of the theoretical results. They are sound and well-written.

---

> ### Author Rebuttal · Authors · 2025-04-01
>
> Thank you for your review! In the following, we respond to a few points raised in your review.
>
> **1. Additional references.**
>
> Thank you for the recommendations! Methods like FUDGE [1] are indeed related to a slight extension to our framework. In particular, the attribute predictor in FUDGE can be thought of as training a process verifier (Definition 3 in our paper) which, instead of outputting binary acceptance / rejection decisions, returns a probability of accepting each prefix. We will cite [1] and references about controllable text generation literature therein, and include a discussion along these lines.
>
> [1] Yang, Kevin, and Dan Klein. FUDGE: Controlled text generation with future discriminators. 2021.
>
> **2. Studying this problem in the settings where proper conditioning is desired.**
>
> We agree that this is an interesting and challenging setting! The main difficulty is understanding what are reasonable assumptions on the power of the process verifier. Proposition 2 suggests that a verifier that solely outputs a binary answer (whether there exists a completion of the prefix in the target domain) is not powerful enough. On the other hand, a simple modification of part 2 of Proposition 1 also shows that a “perfectly calibrated” process verifier (which outputs the conditional probability of the prefix according to the target distribution) can be trivially used to design a linear-time algorithm to sample from the target distribution. Finding reasonable assumptions would involve both theoretical reasoning and incorporating empirical evidence from trained verifiers.
>
> **3. KL divergence between the ground-truth and the trained model in Dyck grammar models.**
>
> In summary: this KL divergence is very small (meaning that the trained model is very well-calibrated against the groundtruth next-token distribution) when completing in-distribution prefixes. For out-of-distribution prefixes, the KL divergence is sometimes large (meaning that the calibration is worse).
>
> More specifically: we define the out-of-distribution prefixes as Dyck_OOD (above Definition 9 in Section 5.1.1 of the paper). Under this setting, a visualization of the calibration results is included in this figure: https://ibb.co/ymSGGsv2.
>
> It can be seen that in this figure, for both the (in-distribution) training set and eval set, the model-predicted log probability of a sequence is generally consistent with the groundtruth log probability. For most sequences generated by the model itself, this consistency also holds, though for some model-generated samples, the model predicts a finite log probability, but the groundtruth log probability should be negative infinite (visualized at -100), because these model-generated samples are not grammatical. For samples generated according to Dyck_OOD, the rightmost subplot of this figure shows that, in many cases, the model can be very miscalibrated.
>
> In calibrated cases, for each prefix, it takes a relatively small number of samples to cover the support for valid next tokens (because vocabulary size is small, and the next token probability is either exactly zero or lower bounded away from zero). In miscalibrated settings, it may take a large number of samples to cover the support for valid next tokens (e.g. when the model predicts an extremely small probability for some correct continuations).

---

### Decision · Program_Chairs · 2025-05-01

**Decision:**

Accept (poster)

**Comment:**

After reviewing the reviewers’ comments and the authors’ detailed rebuttal, I recommend the paper be accepted. The reviewers found the proposed theoretical framework for verifier-assisted language generation to be both rigorous and insightful, significantly advancing understanding of complexity considerations in constrained generation tasks. The paper compellingly demonstrates that verifier-assisted methods can transform computationally difficult problems into tractable ones, with clear and valuable theoretical results. Empirically, the use of tokenwise rejection sampling with backtracking offers strong support for the theoretical claims, showing notable improvements in accuracy, efficiency, and diversity over existing baselines. The authors were proactive in addressing reviewers’ concerns, providing additional experiments, clarifications, and references that strengthened both the theoretical and empirical aspects of the work. Minor concerns include the gap between theoretical assumptions—such as perfect verifiers—and practical implementations using trained, imperfect verifiers, though the authors adequately addressed these with further clarification and experiments. Reviewers also encouraged exploring extensions to probabilistic constraints and real-world tasks beyond structured code generation. Overall, the paper’s theoretical depth, empirical rigor, and thoughtful engagement with reviewer feedback clearly outweigh the minor concerns, and the contribution is recognized as a valuable advancement in the field of constrained language generation.